# CAUSAL-STEER: DISENTANGLED CONTINUOUS STYLE CONTROL WITHOUT PARALLEL CORPORA

**Qingsong Wang[1], Chang Yao[1,*] & Jingyuan Chen[2,*]**

[1]School of Software Technology, Zhejiang University

[2]Zhejiang University

[*]Corresponding authors

{wqsong, changy, jingyuanchen}@zju.edu.cn

## ABSTRACT

Controlling stylistic attributes of Large Language Models (LLMs), such as formality or conceptual complexity, is crucial for effective human-AI interaction. However, current methods often suffer from discreteness, reliance on expensive parallel corpora, and instability, limiting their practical utility. This paper introduces a novel framework for robust activation steering that eliminates the need for parallel corpora, enabling continuous, fine-grained, and linear control over LLM outputs. Our key insight is to reframe Low-Rank Adaptation (LoRA) as a causal intervention tool. By contrasting activations on identical inputs with and without a LoRA perturbation trained via a contrastive objective, we separate the influence of content. To enhance reliability, we introduce a robust aggregation pipeline that uses Principal Component Analysis (PCA) for denoising and the geometric median for centrality estimation, yielding a stable and disentangled style vector. At inference, this vector allows for precise bidirectional control via activation steering with negligible computational overhead. We demonstrate state-of-the-art performance on controlling conceptual complexity, text detoxification, and formality control. Our method not only provides superior control but also generalizes across different models and tasks, and enables simultaneous multi-attribute control. Our code is available at: https://github.com/APTX574/Causal-Steer

## 1 INTRODUCTION

Large Language Models (LLMs) are emerging as powerful general-purpose tools, demonstrating remarkable potential in complex interactive scenarios that demand accurate information delivery (Brown et al., 2020; Touvron et al., 2023a; Lin et al., 2025). Yet, their practical utility extends beyond factual correctness: effective human-AI interaction also requires **fine-grained control over expressive dimensions** of generated output, ranging from linguistic formality to conceptual complexity. Such control is essential for tailoring responses to a user's cognitive state, for example aligning explanations with their Zone of Proximal Development (Vygotsky, 1978). Current mainstream control paradigms, however, remain limited in both precision and stability as show in Figure 1. Prompt engineering (Liu et al., 2023) and instruction fine-tuning (Ouyang et al., 2022) all operate in fundamentally **discrete** control spaces, restricting users to coarse categories (*e.g.*, "beginner" vs. "expert") and producing unpredictable shifts during iterative adjustment. This dis-

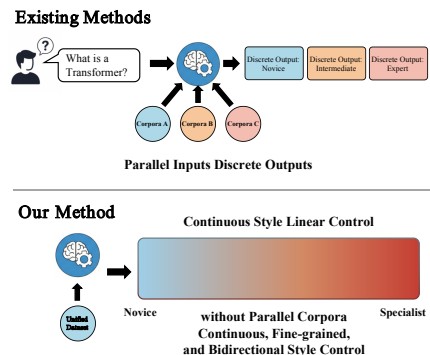

Figure 1: **Comparison of LLM style control**: existing methods require parallel corpora for rigid, discrete control, whereas our method uses non-parallel data to achieve fine-grained, continuous linear control.

creteness is misaligned with the inherently continuous nature of many stylistic spectrums, such as knowledge granularity, ultimately constraining personalization and reducing interaction efficiency.

To achieve continuous adjustment, some studies apply linear interpolation of model parameters (Kangaslahti & Alvarez-Melis, 2025; Ilharco et al., 2022). Using LoRA, a base model is fine-tuned toward different stylistic anchors, and the parameter updates are then interpolated. While this approach forms a continuous control space, it faces key limitations: it often requires parallel data for anchor training and tends to capture corpus-specific artifacts rather than genuine stylistic differences. Consequently, the model merges dataset features instead of learning a transferable representation of style, resulting in outputs that are unstable and difficult to control.

To address these gaps, the emerging paradigm of **activation steering** has attracted increasing attention (Turner et al., 2023a; Zhang et al., 2025). By directly manipulating a model's internal representations in latent space, activation steering offers a principled mechanism for **continuous, fine-grained control** over stylistic attributes. However, the practical effectiveness of this approach hinges on obtaining high-quality steering directions. Existing methods typically rely on *parallel corpora*(Han et al., 2024), pairs of texts aligned in content but differing in style, which are expensive to construct and rarely available for complex dimensions such as conceptual complexity. This dependence introduces two critical challenges. First, the **content pollution problem**: imperfect content alignment causes extracted difference vectors to encode both stylistic and semantic variation, reducing generalizability. Second, the **robustness problem**: even when initial style signals are isolated, steering directions remain vulnerable to noise and outliers, limiting stability across topics.

In this work, we present a novel framework, Causal-Steer, for robust, corpus-free activation steering that enables precise and linear style control. Our key insight is to reframe Low-Rank Adaptation (Hu et al., 2022) (LoRA) as a **causal intervention tool**. By contrasting activations from identical inputs with and without a LoRA perturbation, we isolate the net stylistic effect while circumventing the need for parallel data. To further enhance reliability, we design an aggregation pipeline that applies PCA for denoising and employs robust centrality estimation to derive stable steering vectors resilient to outliers. Using conceptual complexity as a case study, we demonstrate that our method supports continuous bidirectional control and generalizes across tasks and languages.

Our main contributions are as follows:

1. **Linear, Corpus-Free Bidirectional Control.** We extract a style vector from a single, non-parallel dataset (even a single-style dataset) and leverage it to realize fine-grained, linear, and bidirectional control, removing the need for costly parallel corpora.
2. **Robust and Disentangled Representation.** Through PCA-based denoising and robust centrality aggregation, our method suppresses content-related noise and resists outliers, producing a disentangled style vector that remains stable and transferable across diverse settings.
3. **Versatile, State-of-the-Art Performance.** Causal-Steer outperforms prior methods on multiple tasks such as text detoxification and formality control, and consistently generalizes to different domains and languages without additional tuning.

## 2 RELATED WORK

Approaches to style control in LLMs can be broadly categorized into modifications of model parameters (Mañas et al., 2025; Feng et al., 2025) and inference-time activations (Feng et al., 2024; Klein & Nabi, 2024). In the parameter space, Ilharco et al. (2022) introduced "task arithmetic", where vectors derived from fine-tuning are used to edit model capabilities (Akiba et al., 2025). More directly related to style, Dekoninck et al. (2023) demonstrated that interpolating between multiple LoRA adapters, each fine-tuned on different attributes, can effectively control generation style. An alternative, more lightweight paradigm is activation engineering. Pioneering this, Turner et al. (2023a) developed Activation Addition (ActAdd), computing a steering vector from the activation difference of a single pair of contrasting prompts. This concept was refined by Rimsky et al. (2024) with Contrastive Activation Addition (CAA), which averages these differences over a large dataset of pairs for greater stability. In a similar vein, Zhang et al. (2025) proposed Generation with Concept Activation Vectors (GCAV), deriving a controlling vector by training a linear classifier on activations. For more targeted edits, Li et al. (2023) developed Inference-Time Intervention (ITI) to shift activations in specific truth-related attention heads. Finally, Zou et al. (2023) unified these activation-based approaches under the conceptual framework of Representation Engineering.

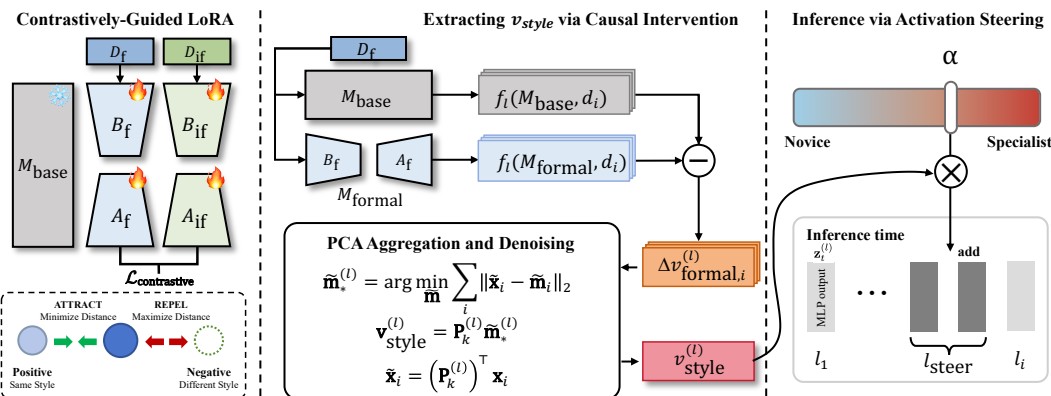

Figure 2: **Framework for extracting and controlling style.** The process is (1) training style-specific LoRAs with a contrastive loss, (2) extracting perturbations caused by the LoRAs and aggregating them into $v_{\text{style}}$, and (3) using this vector for continuous style control at inference. Here, $D_{\text{f}}$ and $D_{\text{if}}$ denote the formal and informal datasets; $B_{\text{f}}$, $A_{\text{f}}$, $B_{\text{if}}$, and $A_{\text{if}}$ are the corresponding LoRA modules; $M_{\text{base}}$ is the base model; $f_l(\cdot)$ is the activation extraction function; $l_{\text{steer}}$ is the steering layer; $\Delta v^{(\ell)}$ is the per-example activation perturbation; and $v_{\text{style}}$ is the aggregated style vector.

## 3 METHODOLOGY

To separate the extracted style vector $\mathbf{v}_{\text{style}}$ from content and mitigate interference from style noise, we aim to achieve fine-grained linear control over text style. Causal-Steer as show in Figure 2, consists of three main stages: 1) We use a contrastively-guided LoRA to introduce a precise, low-rank perturbation to the base model's weights and extract style-specific activation differences from the perturbation. 2) We introduce a robust aggregation technique that combines PCA with a robust centrality estimation to denoise the extracted vectors and identify the core style direction. 3) At inference time, we use the normalized style vector for multi-layer activation steering, enabling continuous and fine-grained style control.

### 3.1 PRELIMINARIES

**Base and Perturbed Models.** $M_{\text{base}}$ is a frozen pretrained language model. $M_{\text{formal}}$ and $M_{\text{informal}}$ are models obtained by applying LoRA-based perturbations to $M_{\text{base}}$, trained on style-specific datasets $D_{\text{formal}}$ and $D_{\text{informal}}$ respectively, using a contrastive loss. Crucially, these datasets need not be parallel, the only requirement is that they exhibit contrasting styles.

**MLP-Layer Activation Extractor.** For a Transformer layer $l$, a model $M$, and a text $d$, we define $f_l(M, d)$ as the mean of the MLP output vectors for all generated tokens, excluding prompt tokens:

$$f_l(M, d) = \frac{1}{T_r} \sum_{t=1}^{T_r} \mathbf{z}_t^{(l)}, \tag{1}$$

where $\mathbf{z}_t^{(l)}$ is the post-MLP hidden state for the $t$-th token, and $T_r$ is the number of generated tokens. This design ensures that the extracted style vector reflects holistic stylistic features throughout the model's output, rather than structural artifacts from the prompt or the partial semantic biases of individual tokens.

### 3.2 EXTRACTING STYLE VECTORS VIA CAUSAL INTERVENTION

LoRA fundamentally applies a precise, low-rank update $\Delta W$ to the base model's weights $W_0$, resulting in fine-tuned weights $W_{\text{style}} = W_0 + \Delta W$. We treat this fine-tuning process as a precise **causal intervention** on the model. Its goal is to elicit specific stylized behaviors while avoiding significant shifts in the core semantic space learned during pretraining. To ensure this perturbation $\Delta W$ effectively separates style from content, we guide its training with a **contrastive learning objective**. This objective directs the weight update towards a subspace that maximizes style separa-

bility. Specifically, the objective function to guide the $M_{\text{formal}}$ perturbation is:

$$\mathcal{L}_{\text{contrastive}} = -\mathbb{E}\left[\log \frac{\exp(\text{sim}(\mathbf{h}_{d_a}, \mathbf{h}_{d_p})/\tau)}{\exp(\text{sim}(\mathbf{h}_{d_a}, \mathbf{h}_{d_p})/\tau) + \sum_{d_n} \exp(\text{sim}(\mathbf{h}_{d_a}, \mathbf{h}_{d_n})/\tau)}\right], \qquad (2)$$

where the anchor $\mathbf{h}_{d_a}$ and positive example $\mathbf{h}_{d_p}$ are from $D_{\text{formal}}$, and negative examples $\mathbf{h}_{d_n}$ are from $D_{\text{informal}}$. This discriminative guidance compels the LoRA update $\Delta W$ to focus on generalizable features that can distinguish the two styles across different content. Consequently, the model must suppress content-related activations, thereby learning a pure and generalizable style representation. We employ a differential method to extract the pure effect of this stylistic perturbation. For a formal sample $d_i$ and layer $l$, the stylistic perturbation vector is defined as:

$$\Delta \mathbf{v}_{\text{formal},i}^{(l)} = f_l(M_{\text{formal}}, d_i) - f_l(M_{\text{base}}, d_i). \qquad (3)$$

The informal difference vector $\Delta \mathbf{v}_{\text{informal},j}^{(l)}$ is calculated analogously.

This differential measurement approach is fundamentally different from naive observational methods. A baseline approach would be to directly obtain the style vector from the difference in activations on the datasets $D_{\text{formal}}$ and $D_{\text{informal}}$ within the base model $M_{\text{base}}$. However, this method assumes that content-related information can be eliminated via vector subtraction, thereby isolating the pure style signal. This assumption holds only under the condition of extremely high content consistency between texts, which in turn necessitates a meticulously crafted parallel corpus. Causal-Steer, in contrast, captures the LoRA-induced perturbation on the very same input $d_i$, elegantly circumventing this reliance on parallel corpora.

Causal-Steer's ability to isolate the stylistic effect hinges on the approximately linear relationship between the weight perturbation $\Delta W$ and the change in activations. We formalize this relationship by considering the activation of layer $l$ as a function of weights $W$ and data $d$, denoted as $\mathbf{h}^{(l)}(W, d)$. A first-order Taylor expansion around the base weights $W_0$ yields:

$$\Delta \mathbf{h}^{(l)}(d) = \mathbf{h}^{(l)}(W_0 + \Delta W, d) - \mathbf{h}^{(l)}(W_0, d) \approx J_{\mathbf{h},W}(W_0, d) \cdot \Delta W, \qquad (4)$$

where $J_{\mathbf{h},W}(W_0, d) = \frac{\partial \mathbf{h}^{(l)}(W,d)}{\partial W}\big|_{W=W_0}$ is the Jacobian matrix mapping perturbations in the weight space to changes in the activation space.

This linear approximation indicates that the $\Delta \mathbf{h}^{(l)}$ we extract can be interpreted as the image of the LoRA-induced weight perturbation $\Delta W$ under this Jacobian mapping. In other words, $\Delta \mathbf{h}^{(l)}$ represents how the stylistic intervention in the weight space manifests within the model's hidden representation space, thus providing a direct and disentangled handle for controlling style.

### 3.3 Robust Aggregation and Denoising

Our objective is to aggregate the collected sample-level difference vectors for each layer into a single, robust style vector $\mathbf{v}_{\text{style}}^{(l)}$. A simple baseline, the arithmetic mean, is suboptimal as it conflates the primary style signal with sample-specific content variations. This indiscriminate averaging introduces noise and fails to isolate the core style direction. To address this, we propose a two-stage strategy that first denoises the vectors to isolate the style subspace and then performs a robust aggregation within that subspace.

**Vector Set Construction and Modeling.** We first construct a unified set of style difference vectors by aligning their directions. Specifically, we negate the vectors derived from the informal style to align them with the formal style direction:

$$X^{(l)} = \{\Delta \mathbf{v}_{\text{formal}}^{(l)}\} \cup \{-\Delta \mathbf{v}_{\text{informal}}^{(l)}\}. \qquad (5)$$

We model each vector $\mathbf{x}_i \in X^{(l)}$ as a composition of a shared, low-dimensional style signal $\mathbf{v}_{\text{style}}^{(l)}$ and high-dimensional, sample-specific content noise $\boldsymbol{\epsilon}_{\text{content},i}$:

$$\mathbf{x}_i = \mathbf{v}_{\text{style}}^{(l)} + \boldsymbol{\epsilon}_{\text{content},i}. \qquad (6)$$

This model posits that the consistent style direction is the primary signal shared across all samples, whereas content-related features manifest as diverse noise.

Table 1: esults on conceptual complexity. Methods with the mean suffix utilize mean token feature extraction for control. Among these, RepE$_{mean}$ failed and is excluded from the ranking, CS denotes our Causal-Steer method, and CS$_{single}$ denotes the variant trained only on the single sided (simple) dataset. For all successful methods, the best results are in **bold** and the second best are underlined.

| Method | Model | Complex | | | | | | | Simple | | | | | | |
|---|---|---|---|---|---|---|---|---|---|---|---|---|---|---|---|
| | | Rel.↑ | Flu.↑ | Acc.↑ | Diff.↑ | F-G.↑ | SMOG.↑ | C-L.↑ | Rel.↑ | Flu.↑ | Acc.↑ | Diff.↓ | F-G.↓ | SMOG.↓ | C-L.↓ |
| CAA | Qwen | 9.56 | 8.71 | 8.19 | 5.37 | 16.28 | 15.57 | 15.64 | 9.57 | 8.51 | 7.58 | 4.18 | 13.64 | 11.19 | 15.71 |
| ITI | 2.5-7B | 9.75 | 9.08 | 8.48 | 5.16 | 16.64 | 15.05 | 15.45 | 9.67 | 8.73 | 7.83 | 4.17 | 13.78 | 17.48 | 17.14 |
| RepE | -Instruct | 9.60 | 8.76 | 8.24 | 5.35 | 16.28 | 14.83 | 14.84 | 9.71 | 8.73 | 8.17 | 4.37 | 14.52 | 17.23 | 17.93 |
| CAA$_{mean}$ | | 9.53 | 8.26 | 7.91 | 8.00 | 19.98 | 19.24 | 29.08 | 9.01 | 8.36 | 6.59 | 2.85 | 11.97 | 12.07 | 10.81 |
| RepE$_{mean}$ | | 9.81 | 9.12 | 8.41 | 4.99 | 16.57 | 15.67 | 17.59 | 9.83 | 9.13 | 8.68 | 4.82 | 14.98 | 15.00 | 14.41 |
| ITI$_{mean}$ | Qwen | **9.78** | **8.41** | 8.30 | 6.97 | 18.25 | 18.03 | 22.06 | 8.94 | 7.87 | 6.71 | 3.12 | 11.92 | 12.52 | 10.80 |
| CLMI | 2.5-7B | 8.92 | 7.97 | 7.60 | 8.36 | 20.67 | **22.49** | 26.91 | **9.50** | **9.06** | 7.64 | 3.47 | 11.38 | 11.37 | 10.02 |
| ReFT | -Instruct | 8.96 | 8.31 | 7.78 | 4.11 | 12.38 | 14.20 | 13.46 | 8.26 | 7.91 | **6.87** | 3.05 | 12.24 | 11.23 | 9.89 |
| CS | | 9.75 | 8.35 | **8.59** | 8.42 | 22.78 | 21.97 | **31.55** | 9.29 | 8.41 | 6.85 | 2.77 | 10.71 | 9.05 | 7.46 |
| CS$_{single}$ | | 9.67 | 8.30 | 8.56 | 8.40 | **23.95** | 21.14 | 27.28 | 9.16 | 8.34 | 6.76 | 2.80 | 10.33 | 10.89 | 9.74 |
| CAA$_{mean}$ | | 8.35 | 6.91 | 5.53 | 6.62 | 20.04 | 19.68 | 21.60 | 6.64 | 4.89 | 4.01 | 1.97 | 12.69 | 10.80 | 9.98 |
| RepE$_{mean}$ | | 9.24 | 8.30 | 6.92 | 4.33 | 14.48 | 15.89 | 16.25 | 9.35 | 8.37 | 6.66 | 4.25 | 13.60 | 14.45 | 15.03 |
| ITI$_{mean}$ | LLaMa | 9.06 | 7.54 | 6.33 | 5.98 | 18.65 | 18.90 | 19.88 | 6.56 | 4.98 | 3.74 | 2.08 | 8.27 | 10.04 | 9.42 |
| CLMI | 3.1-8B | 9.17 | 8.28 | **8.15** | 6.63 | 19.00 | 20.05 | 26.86 | **9.41** | **9.29** | **7.26** | 2.96 | 9.54 | 10.69 | 8.67 |
| ReFT | -Instruct | 7.29 | 7.25 | 6.30 | 3.12 | 3.12 | 11.19 | 8.15 | 8.05 | 8.80 | 5.71 | 2.31 | 7.31 | 9.76 | 7.27 |
| CS | | 9.53 | **8.34** | 7.44 | 7.22 | 20.54 | **22.54** | **29.52** | 6.88 | 6.58 | 4.06 | **1.92** | **6.16** | **8.02** | **6.97** |
| CS$_{single}$ | | **9.59** | 8.31 | 7.94 | **7.81** | **21.63** | 20.17 | 24.64 | 8.04 | 7.61 | 5.18 | 2.16 | 9.04 | 10.22 | 8.79 |

**Denoising via PCA.** Based on our model, we hypothesize that the shared style signal $\mathbf{v}_{style}^{(l)}$ constitutes the principal components with the highest variance in the set $X^{(l)}$. Conversely, the content noise $\epsilon_{content,i}$ is distributed across the remaining components of lower variance. Consequently, we employ PCA to separate the style signal from the content noise. We project each vector $\mathbf{x}_i$ in $X^{(l)}$ onto the subspace spanned by the top $k$ principal components:

$$\tilde{\mathbf{x}}_i = (\mathbf{P}_k^{(l)})^\top \mathbf{x}_i, \tag{7}$$

where the columns of $\mathbf{P}_k^{(l)}$ are the top $k$ eigenvectors of the sample covariance matrix. This projection acts as a filter, preserving the low-dimensional style information while discarding high-dimensional content variations. Empirically, we find that a small value such as $k = 8$ is sufficient to capture the core style variance, supporting our hypothesis that style is a low-dimensional attribute.

**Robust Aggregation with Geometric Median.** While PCA removes structural noise, outliers from atypical samples may persist within the projected style subspace. To mitigate their influence, we perform a robust aggregation using the Geometric Median $\tilde{\mathbf{m}}_*^{(l)}$, which is defined as:

$$\tilde{\mathbf{m}}_*^{(l)} = \arg\min_{\tilde{\mathbf{m}}} \sum_i \|\tilde{\mathbf{x}}_i - \tilde{\mathbf{m}}\|_2. \tag{8}$$

Unlike the arithmetic mean, the geometric median provides a robust measure of centrality that is less sensitive to extreme values. Finally, we project the aggregated vector back to the original activation space to obtain the definitive style vector:

$$\mathbf{v}_{style}^{(l)} = \mathbf{P}_k^{(l)} \tilde{\mathbf{m}}_*^{(l)}. \tag{9}$$

This two-stage procedure yields a robust style vector that generalizes across diverse inputs, enabling reliable and controllable style steering at inference time.

### 3.4 STYLE CONTROL AT INFERENCE VIA ACTIVATION STEERING

A key advantage of Causal-Steer is its ability to enable bidirectional control from a single style vector. This vector can be extracted from a non-parallel corpus representing only the target property (*e.g.*, complex concepts), yet it can steer generation towards or away from that property. We achieve this control at inference time via activation steering, a method that directly modifies a model's internal activations without altering its weights.

Specifically, for a selected set of layers $\mathcal{L}_{steer}$ (see Section 4.3 for selection details), we intervene in the output of the MLP submodule during the generation of each token $t$. We add the pre-computed, normalized style vector $\mathbf{v}_{style}^{(l)}$ to the original activation $\mathbf{z}_t^{(l)}$, scaled by an intensity coefficient $\alpha$:

$$\mathbf{z}_t'^{(l)} = \mathbf{z}_t^{(l)} + \alpha \cdot \frac{\mathbf{v}_{style}^{(l)}}{\|\mathbf{v}_{style}^{(l)}\|_2}. \tag{10}$$

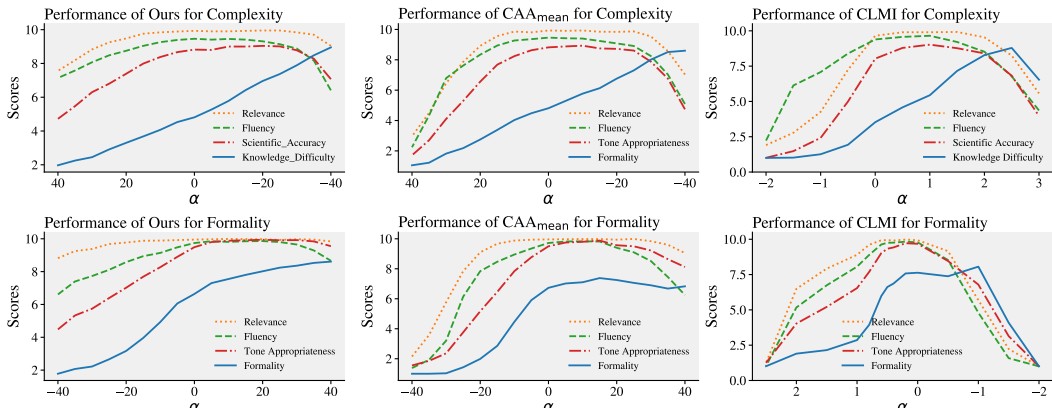

Figure 3: A comparison of the control effectiveness of Causal-Steer against two baselines on Complexity and Formality tasks. The results demonstrate that by adjusting the coefficient $\alpha$, Causal-Steer effectively steers the target style while maintaining high scores for Relevance and Fluency.

Here, $\mathbf{v}_{\text{style}}^{(l)}$ is the robust style vector derived previously. We normalize it to ensure that the intervention strength is determined solely by the tunable scalar $\alpha$.

The coefficient $\alpha$ provides continuous and fine-grained control over the output's complexity level. A positive $\alpha$ guides the model towards the target property (*e.g.*, complex), whereas a negative $\alpha$ steers the activations in the opposite direction (*e.g.*, simple). When $\alpha = 0$, the original model behavior is recovered as the activations remain unchanged. This method transforms complexity control into a single, interpretable parameter. Since style vectors are pre-computed, the intervention is a simple vector addition with negligible computational overhead. This makes the method highly efficient and flexible, allowing for smooth interpolation of concept complexity without any model retraining.

## 4 EXPERIMENTS

**Datasets.** We evaluate Causal-Steer on three stylistic control tasks: conceptual complexity, toxicity detoxification, and formality control. For our primary task of conceptual complexity, we extract style vectors from the **Scale** dataset (Wang et al., 2025) and evaluate on the **ELI5** dataset (Fan et al., 2019). To assess generalization, we test toxicity control using vectors from **APPDIA** (Atwell et al., 2022) on the **RealToxicityPrompts** corpus (Gehman et al.,

Table 2: Ablation study results on the Qwen2.5-7B-Instruct model.

| Method | Complex | | | Simple | | |
|---|---|---|---|---|---|---|
| | Flu. | Acc. | Diff. | Flu. | Acc. | Diff. |
| Causal-Steer | **8.35** | **8.59** | **8.42** | 8.41 | 6.85 | **2.77** |
| -w/o Contrast | 8.24 | 8.50 | 8.24 | 8.31 | 6.87 | 3.01 |
| -w/o Difference | 8.21 | 7.95 | 7.90 | 8.50 | 6.67 | 2.85 |
| -w/o Mean Token | 7.41 | 7.28 | 4.26 | **8.94** | **8.24** | 4.73 |
| -w/o PCA | 8.18 | 8.27 | 8.18 | 7.31 | 5.20 | 2.81 |
| -w/o Mean | 8.28 | 8.48 | 8.24 | 8.36 | 6.61 | 2.82 |

2020), and formality control using data from Zhang et al. (2020) evaluated on **ELI5**. Further details on data preprocessing and statistics are provided in the Appendix D.

**Baselines.** We benchmark Causal-Steer against a comprehensive suite of baselines including Representation Engineering (RepE) (Zou et al., 2023), Contrastive Activation Addition (CAA) (Turner et al., 2023a), Inference-Time Intervention (ITI) (Li et al., 2023), Continuous Language Model Interpolation (CLMI) (Kangaslahti & Alvarez-Melis, 2025), ReFT (Wu et al.), POSPROMPT, Arithmetic (Dekoninck et al., 2023), ActAdd (Turner et al., 2023b), and GCAV-Output (Zhang et al., 2025). Further details on each baseline are provided in the Appendix C.

**Evaluation Metrics.** For conceptual complexity control, we assess Relevance, Fluency, Scientific Accuracy, and Knowledge Difficulty using ChatGPT-4.1 (OpenAI, 2025), supplemented by the automated Flesch Grade Level (Flesch, 2007), SMOG (Mc Laughlin, 1969), Coleman-Liau (Coleman & Liau, 1975), and human evaluation in the Appendix E. For formality control, we evaluate Tone Appropriateness, Relevance, Fluency, and Formality using GPT-4.1. We also use the popular s-

nlp/roberta-base-formality-ranker model[1] from Hugging Face as the text formality style prediction model. This is a binary classification model, and we use the probability value as the text formal style strength. Finally, detoxification performance is measured by the maximum toxicity score from the Perspective API[2]. Full evaluation prompts are detailed in the Appendix G.

For the ChatGPT-4.1 metrics, we conducted three measurements and reported their average. Because the ChatGPT-4.1 scores were highly consistent, the standard deviations across the three runs were very small. Due to space constraints, we report only the average values and omit the standard deviations. All scores range from 1 to 10.

## 4.1 MAIN RESULTS ON CONCEPTUAL COMPLEXITY CONTROL

Table 1 presents the quantitative results for controlling conceptual complexity. Causal-Steer significantly outperforms most baselines across both the Qwen2.5-7B-Instruct(Qwen et al., 2024) and LLaMa3.1-8B-Instruct(Grattafiori et al., 2024). The framework demonstrates strong style control while maintaining high generation quality. When steering towards "Complex", Causal-Steer exhibits superior control. On Qwen2.5-7B, for instance, it achieves the highest difficulty score (8.42) and Flesch grade (22.78), producing conceptually advanced content without sacrificing relevance or fluency. Conversely, for the "Simple" condition, Causal-Steer consistently records the lowest difficulty and Flesch grade scores. These results confirm our vector's capacity for precise bidirectional steering.

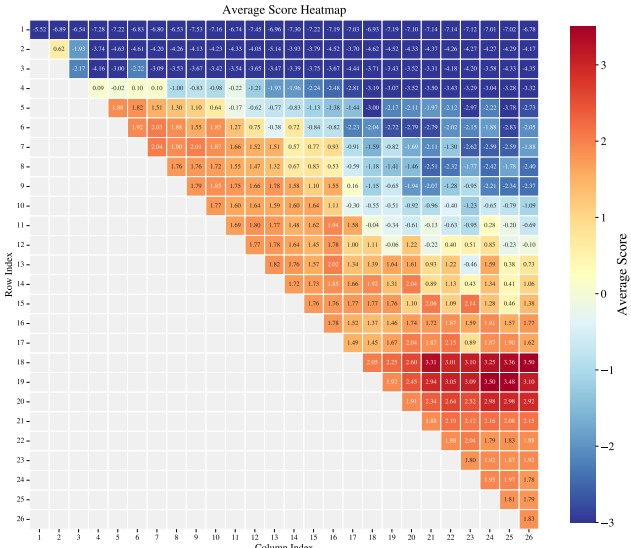

Figure 4: **Heatmap of average style control scores across layer blocks.** The y-axis and x-axis represent the start and end indices for the intervention block, respectively. Scores are averaged over multiple runs.

The analysis of the baselines reveals the importance of our activation strategy. Methods like the original CAA, ITI, and RepE rely on activations from the final token. This approach necessitates high-quality parallel corpora to be effective. As shown in the table, adapting CAA to use mean activations (CAA$_{mean}$) substantially enhances its control capabilities. In contrast, RepE$_{mean}$ still fails. We attribute this failure to its use of PCA for analyzing activations, a technique that requires a much larger dataset to identify a meaningful style direction. This hypothesis is supported by our generalization experiments, where its performance improves on the larger Formality dataset (16,000 examples) (Zhang et al., 2020).

Although CLMI achieves commendable results in style control, its primary limitation is a failure to disentangle style from content during fine-tuning. Consequently, the model learns to associate the target style with specific content from its training data, leading to the generation of content artifacts. CLMI's outputs are consistently shorter (approx. 100 tokens) than those of other methods (approx. 500 tokens), mirroring its training data in both length and structure. This highlights that Causal-Steer succeeds by isolating a pure style vector. This isolation enables robust, generalizable control, which is unachievable by other methods that suffer from severe content-style entanglement.

---

[1]https://huggingface.co/s-nlp/roberta-base-formality-ranker

[2]https://perspectiveapi.com

## 4.2 SINGLE-SOURCE BIDIRECTIONAL CONTROL

Beyond the standard setting with two style-specific LoRA adapters, Causal-Steer also supports bidirectional control when both the adapter and the steering vector are learned from a single-sided corpus. To verify this, we construct a variant, denoted **CS$_{single}$**, that uses only the *Simple* subset of the Scale dataset. In this variant, we remove the contrastive loss used in Section 3 and fine-tune a *single* LoRA on Simple answers with the standard supervised objective. We then apply the same causal-intervention procedure to extract one style vector. At inference time, we use a positive steering coefficient $\alpha > 0$ for the learned style and a negative coefficient $\alpha < 0$ to steer in the opposite (more complex) direction. As shown in Table 1, **CS$_{single}$** achieves remarkable performance, effectively demonstrating robust bidirectional control even when trained on single-sided data.

We further compare this single-source setting against ReFT. While ReFT performs well when steering in the learned direction (positive scaling), it fails significantly when we attempt bidirectional control. Specifically, applying a negative coefficient for ReFT does not yield the opposite style; instead, it leads to severe generation artifacts, such as repetition and hallucinations, without successfully shifting the style. We hypothesize that this failure stems from the nature of the learned representation: ReFT likely learns a vector representing the *residual* between the base model and the dataset outputs, rather than an intrinsic *style vector*. Consequently, simply reversing this residual is semantically meaningless, causing model collapse. In contrast, our method explicitly learns the style vector itself, thereby enabling stable and valid bidirectional control.

## 4.3 ANALYSIS OF THE CONTROL MECHANISM

To better understand the properties of Causal-Steer, we conduct a deeper analysis of its control linearity and layer-wise sensitivity.

**Linear and Stable Stylistic Control.** A core objective of Causal-Steer is to enable continuous and predictable control over style. Figure 3 evaluates this by plotting performance metrics against the steering coefficient $\alpha$. The results for Causal-Steer demonstrate a strong, approximately linear relationship between $\alpha$ and the target style metrics, Formality and Knowledge Difficulty. As $\alpha$ increases, the intended stylistic intensity grows predictably. Critically, this control is achieved with minimal impact on generation quality. Key met-

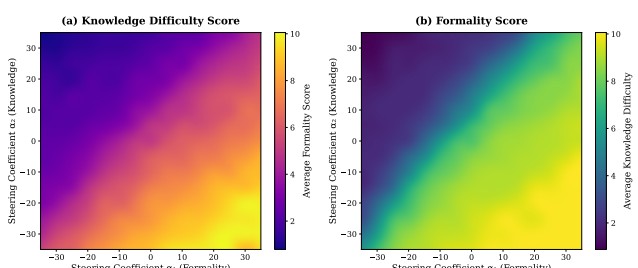

Figure 5: **Two-Dimensional Stylistic Control.** Model output scores for (a) Knowledge Difficulty and (b) Formality under simultaneous steering.

rics such as Relevance and Fluency remain high across a wide operational range of $\alpha$ values, showing degradation only at extreme settings. This stability highlights the effectiveness of our robust aggregation pipeline.

The CLMI reveals a more fundamental limitation. Its control is effectively confined to the spectrum between its two training endpoints, corresponding to a safe operational range for $\alpha$ between 0 and 1. Because this method functions by interpolating between the weights of two fine-tuned models, it cannot extrapolate beyond the styles observed during training. Attempting to push the model outside this bounded range, for instance, by setting $\alpha$ to values like -1 or 2, results in a catastrophic collapse in generation quality, causing all performance metrics to plummet. This means the method cannot generate content that is, for example, more formal than the examples in its formal dataset without sacrificing coherence. This fundamentally restricts its utility to a narrow style range and prevents true creative or intensified style generation. The comparison confirms the superiority of Causal-Steer in creating a genuinely continuous, wide-ranging, and robust control mechanism.

**Identifying Optimal Layers for Intervention.** We investigate which layers are most influential for style control to inform an optimal intervention strategy. Figure 6 visualizes the average style score achieved by applying our steering vector across different continuous blocks of layers, defined

by a start and end index in Qwen2.5-7B-Instruct. When calculating the average score, we applied a penalty for outputs that collapsed into repetitive, meaningless content. The heatmap reveals a clear pattern: the most effective control (indicated by the highest positive scores in orange and red) is concentrated in the mid-to-late layers of the model. Specifically, interventions starting around layer 18 and extending to approximately layer 23 yield the strongest stylistic effect. Intervening in the initial layers (1-10) proves far less effective and can even be detrimental to the output. This empirical result aligns with the prevailing hypothesis that later transformer layers encode more abstract semantic and stylistic information (Zhang et al., 2025). Based on this analysis, we apply steering to layers 18 through 23 in our main experiments, as this range provides a robust and powerful locus for style manipulation.

## 4.4 ABLATION STUDIES

We conduct an ablation study to validate the necessity of each component in Causal-Steer. As shown in Table 2, every component is critical for optimal performance.

Removing the contrastive learning objective ("-w/o Contrast") causes a notable decline in control effectiveness. Forgoing the differential acti-

Table 3: Performance of different methods on Formal and Informal datasets.

| Method | Formal | | | | | Informal | | | | |
|---|---|---|---|---|---|---|---|---|---|---|
| | Rel.↑ | Flu.↑ | Tone.↑ | For.↑ | P.F.↑ | Rel.↑ | Flu.↑ | Tone.↑ | For.↓ | P.F.↓ |
| CAA$_{mean}$ | 9.05 | 6.25 | 8.12 | 6.83 | 0.91 | 9.28 | 8.08 | 5.65 | 2.30 | 0.32 |
| RepE$_{mean}$ | 9.79 | 8.47 | 8.82 | 6.38 | 0.96 | 8.90 | 6.68 | 5.59 | 2.80 | 0.72 |
| ITI$_{mean}$ | 9.73 | 8.26 | 8.29 | 5.86 | 0.91 | 7.54 | 5.12 | 4.71 | 2.71 | 0.78 |
| CLMI | 9.80 | 9.37 | 9.41 | 7.58 | 0.88 | 8.82 | 7.70 | 5.94 | 2.40 | 0.36 |
| CS | 9.84 | 8.66 | 9.55 | 8.61 | 0.98 | 9.38 | 7.73 | 5.74 | 2.22 | 0.20 |

vation ("-w/o Difference") also measurably weakens control. Both results validate our causal intervention strategy. The aggregation pipeline is equally important; removing PCA ("-w/o PCA") or using a simple arithmetic mean ("-w/o Mean") degrades performance by failing to properly denoise the style signal. Finally, using only the last token's representation ("-w/o Mean Token") instead of averaging across all tokens severely diminishes control intensity, underscoring the need for a holistic style signal. These findings confirm that all components contribute synergistically to the framework's effectiveness.

## 4.5 MULTI-ATTRIBUTE CONTROL

Causal-Steer's capabilities extend beyond single-attribute manipulation to natural, multi-dimensional style control. This is theoretically grounded in the sparse nature of high-dimensional vector spaces, which permits the linear superposition of multiple style vectors (Liang et al., 2024). By simply adding the vectors for different attributes, we can achieve simultaneous and composable control over the generation process.

Figure 7 provides an empirical demonstration of this principle, where we jointly steer for Formality and Conceptual Complexity . The figure presents two heatmaps showing the resulting style scores as a function of the formality coefficient $\alpha_1$ (horizontal axis) and the knowledge coefficient $\alpha_2$ (vertical axis). Specifically, Subfigure (a) illustrates the model's output **Knowledge Difficulty Score**, while Subfigure (b) shows the generated **Formality Score**. The heatmaps demonstrate that as the control signals $\alpha_1$ and $\alpha_2$ are varied, the corresponding style scores undergo a **smooth and continuous transition**. Furthermore, the plots reveal a positive correlation between the two attributes. Intensifying one style's control signal produces a corresponding change in the score for the other style. For example, increasing the conceptual complexity signal ($\alpha_2$) also tends to increase the formality score, and vice versa. This observation highlights the inherent semantic interplay between advanced conceptual content and formal language. The smooth and predictable gradients across both heatmaps provide strong evidence that Causal-Steer enables fine-grained and continuous control over multiple stylistic dimensions, while effectively capturing their natural associations.

## 4.6 GENERALIZATION TO OTHER CONTROL TASKS

To assess the generalizability of Causal-Steer beyond conceptual complexity, we evaluated its performance on two distinct stylistic control tasks: formality control and text detoxification.

Table 3 presents the results for Formality task. Causal-Steer demonstrates superior control over the target style, achieving the highest formality score (8.61) when steering towards formal style and

one of the lowest scores (2.22) when steering towards informal style. Crucially, this high degree of stylistic control is attained while simultaneously yielding the highest scores for relevance and scientific accuracy, indicating that Causal-Steer effectively modulates style without degrading content quality.

Furthermore, we applied Causal-Steer to the critical safety task of detoxification with Llama-2-7b-chat (Touvron et al., 2023b), with results shown in Table 4. Causal-Steer significantly outperforms all baselines, achieving the lowest toxicity scores on both the toxic and random test sets, where both evaluations involve one-way detoxification rather than the bidirectional control used in our other experiments. This detoxification performance is achieved while maintaining a competitive perplexity (PPL), suggesting that Causal-Steer ef-

Table 4: Comparison of toxicity control methods on two test sets. Toxicity (lower is better) and perplexity (PPL, lower is better) are reported.

| Model | Toxicity$_{\text{toxic}}$ | | Toxicity$_{\text{random}}$ | |
|---|---|---|---|---|
| | Toxicity ↓ | PPL ↓ | Toxicity ↓ | PPL ↓ |
| Baseline | 0.1807 | 13.71 | 0.0956 | 19.23 |
| POSPROMPT | 0.1913 | 59.98 | 0.1008 | 18.32 |
| ActAdd | 0.1620 | 34.08 | 0.0852 | 12.61 |
| Arithmetic | 0.1625 | **6.84** | 0.0816 | 7.34 |
| GCAV-Output | 0.0879 | 21.29 | 0.0622 | **6.08** |
| Causal-Steer | **0.0609** | 12.95 | **0.0520** | 8.18 |

fectively mitigates toxicity without severely compromising the model's linguistic coherence. These experiments collectively demonstrate that our proposed vector extraction and steering mechanism provides a versatile and effective solution for a wide range of attribute control problems. We also tested Causal-Steer cross-lingually, detailed cases are in Appendix J.

Table 5: Comparison of Model Responses

| $\alpha$ | **Question:** Do magnetic and/or electric field have any influence on time and space? |
|---|---|
| 10 | Yes,..., Electric and magnetic fields do not directly change spacetime, but they can affect how particles move, which influences how time and distance are measured in practice... |
| -5 | Yes,..., In general relativity, electromagnetic fields contribute to spacetime curvature through their energy, meaning that strong fields can slightly alter the geometry of spacetime... |
| -20 | Yes,..., Within the framework of Einstein's general relativity, the interaction between electromagnetism and gravity is formally described by the **Einstein–Maxwell equations**, where the stress–energy tensor of the electromagnetic field directly enters Einstein's field equations to determine spacetime curvature... |

## 4.7 CASE STUDY

Table 5 illustrates representative responses under different control strengths $\alpha$. As $\alpha$ decreases, the generated answers gradually shift from intuitive and accessible descriptions to more theoretical and domain-specific explanations, ranging from practical comments on measurement effects ($\alpha = 10$) to discussions of spacetime curvature ($\alpha = -5$), and finally to highly academic references such as the Einstein–Maxwell equations ($\alpha = -20$). This demonstrates that $\alpha$ serves as a continuous control knob, enabling smooth adjustment of responses along a spectrum from layperson-friendly to technical. For brevity, we only present a condensed example here, while the complete outputs and additional cases are provided in the Appendix H.

## 5 CONCLUSION

We introduced **Causal-Steer**, a novel framework for fine-grained style control in LLMs that removes the dependency on parallel corpora by leveraging LoRA as a causal intervention to disentangle style vectors. Supported by a robust aggregation pipeline, our method successfully isolates a pure style signal from non-parallel data. Experimental results demonstrate that Causal-Steer enables linear, bidirectional, and compositional control over diverse stylistic attributes such as conceptual complexity and formality, while preserving generation quality and exhibiting strong cross-lingual adaptability. These findings highlight the practicality and effectiveness of our approach, offering a continuous and robust paradigm for adaptive and steerable language models beyond discrete control.

ACKNOWLEDGMENTS

This research was supported by the National Natural Science Foundation of China (No. 62507040), the Ningbo "Yongjiang Talent Program" Youth Innovation Project (No. 2024A-156-G), and the "Pioneer" and "Leading Goose" R&D Program of Zhejiang (No. 2025C02022).

ETHICS STATEMENT

Our research fully adheres to ethical guidelines for responsible AI and machine learning research. All datasets used in our experiments are publicly available datasets. No proprietary, sensitive, or private data was used in this study.

In addition to computational experiments, we conducted a small-scale human study to assess text style. Participation was voluntary, and all participants provided informed consent before taking part. The study was reviewed and approved under our institution's internal review procedures, and participants' privacy and confidentiality were fully protected. All collected data was anonymized and used exclusively for research purposes.

We acknowledge that responsible research entails careful consideration of potential harms and social impacts. Our work does not involve manipulative, discriminatory, or unsafe practices, and we have designed our experiments to minimize any negative consequences to participants and the broader community.

REPRODUCIBILITY STATEMENT

We have taken several measures to ensure the reproducibility of our work. First, detailed descriptions of our methods, model architectures, and algorithms are provided in the main text, and the corresponding code is made available through the anonymous repository `https://anonymous.4open.science/r/cs-01C1`, which is also referenced in the supplementary materials. Second, all datasets used in our experiments are publicly available and included in the anonymous repository, along with complete data processing steps and instructions necessary to replicate our experiments. Third, in the appendix, we provide detailed explanations of the experimental setup, hyperparameters, evaluation protocols, including prompt design and review procedures, to allow accurate replication of our computational and human-in-the-loop experiments. Together, these resources provide all information necessary for reproducing our results.

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

## A  STATEMENT ON THE USE OF LLMS

We report the use of a large language model in the preparation of this manuscript. Specifically, we used Gemini 2.5 (Gemini Team, 2025) to identify and correct grammatical errors and to improve the overall readability of the text. The authors retained full responsibility for all content, and the final version of the paper reflects our own edits and revisions.

## B  DETAILS OF EXPERIMENTAL CONFIGURATION

To ensure the reproducibility of our results, we provide a detailed description of the experimental configuration used for training, inference, and evaluation. All experiments were conducted on four NVIDIA A100 GPUs with 80GB memory each. Our model was trained for a total of 2 epochs with a per-device batch size of 8, a learning rate of $1 \times 10^{-4}$, gradient accumulation steps set to 1, and a maximum sequence length of 512 tokens. For parameter-efficient fine-tuning, we employed Low-Rank Adaptation (LoRA) with a rank $r = 8$, scaling factor $\alpha = 16$, and a dropout rate of 0.05. The `PEFT` library was used to automatically identify and adapt the appropriate linear layers within the model. In addition, a contrastive learning objective was incorporated with the contrastive loss weight $\lambda = 0.05$. For inference, we used deterministic decoding (greedy search without sampling) with a maximum generation length of 512 tokens. During evaluation with GPT-based scoring, the temperature was set to 0.2 to ensure stability and consistency of judgments.

## C  DETAILS OF BASELINES

We benchmark our method against a comprehensive suite of baselines for controllable generation, categorized into prompt-based and activation-based approaches. For our primary task of conceptual complexity control, we include zero-shot instructional prompting and three prominent activation steering methods: Representation Engineering (RepE) (Zou et al., 2023), Contrastive Activation Addition (CAA) (Turner et al., 2023a), and Inference-Time Intervention (ITI) (Li et al., 2023). Distinct from these activation-based techniques, we also include Continuous Language Model Interpolation(CLMI) (Kangaslahti & Alvarez-Melis, 2025) for our main experiment, which achieves control by interpolating the weights of Low-Rank Adapters (LoRA) rather than steering activations. For the activation-based methods, we evaluate both the standard approach using the final token's hidden state and a stronger variant we implement that averages hidden states across all tokens (**-avg**). For the toxicity detoxification task, we include additional specialized baselines such as the prompt-based POSPROMPT and activation steering methods including Arithmetic (Dekoninck et al., 2023), ActAdd (Turner et al., 2023b), and GCAV-Output (Zhang et al., 2025).

## D  DETAILS OF DATASETS

We evaluate our method on a primary task of controlling conceptual complexity and assess its generalizability on two additional stylistic dimensions: toxicity detoxification and formality control.

For our primary task, conceptual complexity, we extract the corresponding style vector from the **Scale**[3], a question-answering corpus with graded difficulty levels. We then evaluate the model's

---

[3]https://huggingface.co/wa57/Scale

ability to modulate content difficulty on a test set of 100 questions randomly sampled from the **ELI5** dataset (Fan et al., 2019).

To demonstrate the versatility of our method, we conduct two generalization experiments. For toxicity control, we derive the style vector from the **APPDIA** dataset (Atwell et al., 2022), which provides aligned toxic and detoxified sentence pairs. We evaluate its performance following the setup of Zhang et al. (2025) on two subsets of the **RealToxicityPrompts** corpus (Gehman et al., 2020): one with highly toxic prompts and another with randomly sampled ones. For formality control, we extract the style vector from 16,000 sentence pairs randomly sampled from the parallel corpus of Zhang et al. (2020).

Since both the APPDIA and formality datasets are originally composed of parallel sentences rather than question–answer pairs, we adapt them to the instruction-following setting: we use **GPT-4.1** to generate corresponding questions $q$ for each pair and slightly modify the answers to better align with instruct-style training. The exact prompting procedure is detailed in Appendix G.1, and the adapted datasets are provided both in the appendix and via our anonymous repository.

## E  HUMAN EVALUATION

To better assess the relative performance of our method, we conducted a human evaluation using a Best-Worst Scaling methodology. This approach reduces annotator bias compared to direct scoring. For each of the 50 test questions, 3 annotators were shown the *Simple* and *Complex* outputs from our method, CAA$_{mean}$, and CLMI in a randomized, blind setting. For each set of three, they were asked to select the single "Best" and "Worst" response based on a holistic judgment of style appropriateness and overall quality (*e.g.*, fluency, coherence).

Table 6: Human evaluation results based on Best-Worst ranking. We report the percentage of times each method was chosen as Best, Middle, or Worst by 3 annotators across 50 questions. A higher 'Best' percentage indicates stronger preference.

| Method | Complex Style (%) | | | Simple Style (%) | | |
|---|---|---|---|---|---|---|
| | Best | Middle | Worst | Best | Middle | Worst |
| CLMI | 11.3 | 25.3 | 63.3 | 14.7 | 28.0 | 57.3 |
| CAA$_{mean}$ | 23.3 | 41.3 | 35.3 | 26.0 | 45.3 | 28.7 |
| Ours | 65.3 | 33.3 | 1.3 | 59.3 | 26.7 | 14.0 |

The results, summarized in Table 6, show a strong preference for our method. For the *Complex* style, our model was chosen as "Best" in a decisive **65.3%** of cases, significantly outperforming CAA$_{mean}$ (23.3%) and CLMI (11.3%).

A similar trend was observed for the *Simple* style, where our method was again preferred as "Best" in **59.3%** of evaluations. This confirms its robust bidirectional control. The low percentage of our method being selected as "Worst" (1.3% for Complex and 14.0% for Simple) further underscores its stability and reliability. In contrast, CLMI was rated "Worst" in the majority of cases (63.3% for Complex), highlighting its limitations in achieving effective style control. These findings from our comparative human evaluation strongly validate the superiority of our proposed framework.

## F  COMPUTATIONAL OVERHEAD

The primary computational cost of our framework is incurred during the offline style vector extraction phase. In contrast, the overhead during online inference is negligible. As shown in Table 7, the generation speed, measured in tokens per millisecond, remains nearly identical to the baseline even when applying multiple steering vectors simultaneously. The control intensity ($\alpha$) has no impact on this speed, as the underlying operation is a simple vector addition. This efficient design, which concentrates computational effort into a one-time offline process, makes our method highly practical for real-time applications.

Table 7: Inference speed in tokens per millisecond (tokens/ms). Higher values are better. Our method maintains a speed comparable to the baseline.

| Configuration | Speed (tokens/ms) |
|---|---|
| Baseline | 1.17 |
| 1 vector, $\alpha = 20$ | 1.13 |
| 1 vector, $\alpha = 30$ | 1.16 |
| 2 vectors, $\alpha = 20$ | 1.12 |
| 2 vectors, $\alpha = 30$ | 1.04 |

## G  PROMPTS

### G.1  PROMPTS FOR DATASETS REWRITE

Transform the given pair of parallel sentences (e.g., formalinformal, toxicdetoxified) into a unified QA format.

Steps

1. Generate one natural and coherent question (Q) that could reasonably elicit both given answers.

2.Produce two answers (A1 and A2) corresponding to the styles in the original parallel data:
  – If the pair is formal vs. informal, ensure that A1 is informal and A2 is formal.
  – If the pair is toxic vs. detoxified, ensure that A1 retains the toxic style while A2 provides the detoxified version.

3. You may expand the answers with additional content to increase richness, but keep both answers semantically consistent with each other.

4. Always preserve the intended style (informal vs. formal; toxic vs. detoxified).

### G.2  PROMPTS FOR CONCEPTUAL COMPLEXITY TASK

You are an expert evaluator for natural language generation quality.

Evaluate the following text based on four dimensions. Focus especially on conceptual complexity.

1. Relevance: Does the response stay on topic and directly address the question or topic without irrelevant content?

2. Fluency: Is the language clear, coherent, and easy to read, with proper grammar and logical flow?

3. Scientific Accuracy: Are the knowledge points and explanations factually correct and reliable based on authoritative sources?

4. Knowledge Difficulty: Assess whether the knowledge level matches the expected difficulty.
    – Higher scores: Highly technical or research–level content suitable for graduate–level understanding.
    – Lower scores: Simple, introductory, or popular–science level explanations.

Give each dimension a score from 1 (poor) to 10 (excellent).

### G.3  PROMPTS FOR FORMALITY TASK

You are an expert evaluator for natural language generation quality.

Evaluate the following text based on four dimensions. Focus especially on formality.

1. Relevance: Does the response stay on topic and directly address the question or topic without irrelevant content?

2. Fluency: Is the language clear, coherent, and easy to read, with proper grammar and logical flow?

3. Tone Appropriateness: Is the tone of the response (e.g., professional, casual, empathetic, neutral) suitable for the prompt's context and implied audience?

4. Formality: Assess the text's level of formality based on word choice, sentence structure, and adherence to stylistic conventions.(Assign a score of 5 for standard model output.)
  – Higher scores: Highly formal language, sophisticated vocabulary, complex sentence structures, and an objective tone. Avoids slang, contractions, and colloquialisms.
  – Lower scores: Highly informal or conversational language. May use slang, contractions, simple sentence structures, and a personal or subjective tone.

Give each dimension a score from 1 (poor) to 10 (excellent).

# H    CASE OF CONCEPTUAL COMPLEXITY TASK

**Question:** Do magnetic and/or electric fields have any influence on time and space?

$\alpha = 10$ (Simple)

Yes, both the electric and magnetic fields can affect how we measure time and distance in certain situations, especially when dealing with very fast speeds close to light or strong gravitational fields.
1.**Electric Field**: In general relativity, an electric charge at rest creates no direct effect on spacetime that would change our usual way of measuring distances and times (like using clocks). However, if you consider charged particles moving through an electromagnetic field, it's important to remember that they experience forces from those fields which could make their paths twist around as seen from other frames of reference. But this doesn't directly mean there is a change in the measurement of time and space for them; rather, it means that things get more complicated because now you need to take into account not just gravity but also these extra forces coming from the electric charges.
2.**Magnetic Field**: Similar to the case with electric fields, a single magnet sitting still wont cause changes in measurements of time and space unless something else interacts with its field. When considering objects moving near each other under the influence of magnets, like two wires carrying current, then yes, the movement of one wire affects another due to the changing magnetic field, making everything move differently than expected without taking the magnetic force into consideration. This makes the path taken look different depending where you're standing  kind of like seeing someone walk diagonally instead of straight ahead while riding on a merry–go–round!
3.**Spacetime Curvature Due To EM Energy**: Just like mass curves spacetime according to General Relativity, energy carried by photons (light) does so too. So having lots of electricity and magnetism swirling about will curve spacetime slightly, much like putting a heavy ball down onto a trampoline causes it to bend downwards. It might be tiny bending compared to planets and stars, but it exists!
In summary, while individual static electric and magnetic fields don't usually warp time and space themselves, once you start talking about moving charges.

$\alpha = -5$ (Medium)

Yes, both the electromagnetic fields (electric and magnetic) can indeed affect spacetime according to modern physics theories, particularly general relativity.
In Albert Einstein's theory of General Relativity, gravity is described as curvature in spacetime caused by mass and energy. This means that not only does massive matter curve spacetime but also strong or dynamic electromagnetic fields can contribute to this curvature under certain conditions. Specifically:
1. **Einstein–Maxwell Equations**: These equations combine Maxwells equations for electromagnetism with Einsteins field equations from general relativity. They describe how an electromagnetic field affects the geometry of spacetime through its stress–energy tensor term.
2. **Gravitomagnetic Effects**: In extreme scenarios involving very high currents or intense magnetic fields, there could be significant gravitomagnetic effects predicted by these unified models. However, such effects would typically require extremely large–scale phenomena beyond what we encounter in everyday situations.
3. **Quantum Gravity Theories**: At quantum scales, where gravitational forces become comparable to other fundamental interactions like those mediated by the electromagnetic force, more speculative theoretical frameworks may come into play. Some proposals within loop quantum gravity or string theory suggest that even at microscopic levels, electromagnetic fields might influence spacetime structure indirectly via their contributions to the overall quantum state of spacetime.
4. **Experimental Observations**: While direct experimental evidence supporting these predictions remains limited due to technological constraints, indirect observations and precision measurements continue to refine our understanding of how strongly coupled systems behave near black holes or neutron starsenvironments characterized by highly relativistic and possibly magnetized conditions.
It's important to note that while theoretically possible, the practical implications of electromagnetic influences on spacetime remain largely confined to astrophysical contexts rather than macroscopic terrestrial environments. Nonetheless, ongoing research continues to explore these fascinating intersections between electrodynamics and gravitational physics.

$\alpha = -20$ (Complex)

> Yes, both electromagnetic fields (which encompass the electric and magnetic fields) can indeed exert influences on spacetime within certain theoretical frameworks of modern physics.
> ### Electromagnetic Fields in General Relativity
> In the context of general relativity, formulated by Albert Einstein, gravitational effects are described through the curvature of spacetime. However, this framework primarily accounts for mass−energy contributions to such curvature. Recent interpretations and extensions of these theories incorporate electrodynamics into considerations of spacetime structure:
> 1. **Quantum Gravity Theories**: These speculative frameworks attempt to reconcile quantum mechanics with general relativity. In some models, quantized electromagnetic interactions could contribute non−trivially to spacetime metrics at microscopic scales.
> 2. **Non−Linear Electrodynamics**: Certain modifications or nonlinear formulations of classical electromagnetism predict deviations from Maxwellian predictions under extreme conditions relevant to high−field regimes near black holes or compact astrophysical objects.
> ### Quantum Electrodynamic Effects
> At the microscale, phenomena governed by quantum electrodynamics (QED), which is an extension of classical electrodynamics incorporating principles of quantum mechanics, exhibit significant implications:
> 1. **Casimir Effect**: This phenomenon arises due to quantum fluctuations in vacuum states between two closely spaced conductive surfaces. It involves spatial configurations influenced by electromagnetic boundary conditions, manifesting as measurable physical forces.
> 2. **Vacuum Polarization**: Within QED, virtual particle pairs (electrons and positrons) transiently interact with intense electromagnetic fields, leading to observable macroscopic consequences via renormalization parameters.
> ### Lorentz Violations and Anisotropic Spacetimes
> Theoretical explorations often invoke violations of local Lorentz symmetryspatial isotropy and temporal homogeneity postulated foundational tenets of special relativistic formalismsto model emergent phenomenological descriptions consistent with experimental data across various energy scales.

## I    CASE OF FORMALITY TASK

**Question:** why do most bipedal robots always keep their knees bent a bit when standing?

$\alpha = -25$ (Informal)

> Most bipedal robots keep their knees slightly bent for several reasons:
>
> 1. **Stability**: Keeping the legs in a slight bend helps with balance and stability. When the robot is standing still or walking, this position makes it easier to stay upright without falling over.
> 2. **Energy Efficiency**: A little bend in the knee can help save energy because its not as hard on the motors that move the leg parts (like servos). It's like how you might bend your knees before jumping it uses less power but gives more push−off force.
> 3. **Shock Absorption**: Just like humans, bending at the knees can act as a shock absorber during steps or jumps. This means the robot doesnt have to work so hard to handle bumps or uneven ground.
> 4. **Flexibility \& Range of Motion**: Bending the knees keeps them from getting stuck straight up or down all the time. This lets the robot be more flexible about where its feet go next, which is important if it needs to walk around stuff or change directions fast.
> 5. **Less Stress On Joints And Bones** : In real life, keeping joints loose instead of locked up stops things from breaking under pressure. Same goes for these bots – they don't want metal bones snapping!
>
> So basically, just like we use our bodies' bends to feel comfy and strong while standing, robots get way into those same tricks too!

$\alpha = -10$

---

Most bipedal robots maintain a slight bend in their knees while standing for several reasons:

1. **Stability**: Keeping the legs at an angle (bent) helps distribute weight more evenly and provides better stability against tipping over. This is similar to how humans stand with slightly bent knees.
2. **Energy Efficiency**: A small amount of knee flexion can help store potential energy as elastic strain in the leg muscles or tendons. When walking or running, this stored energy can be released to assist in propelling the robot forward, making it more efficient.
3. **Shock Absorption**: The slight bend acts like a shock absorber during movement. It reduces impact forces on both joints and other parts of the body, which could otherwise cause damage from repeated hard landings.
4. **Joint Health**: Maintaining some degree of joint flexibility prevents stiffness that might occur if they were locked straight all the time. Stiffness increases wear and tear on mechanical components.
5. **Control Precision**: In robotics, keeping certain angles allows engineers to fine-tune control algorithms because there's less chance of hitting physical limits compared to having fully extended limbs where you'd hit "hard stops."
6. **Humanoid Design Consistency**: For humanoid robots designed to mimic human movements, maintaining a natural stance makes them look more lifelike and easier to interact with since people expect others to have naturally curved postures rather than rigidly upright ones.

Overall, these factors contribute to improving performance, longevity, and user interaction for bipedal robots!

---

$\alpha = 10$

---

Most bipedal robots maintain a slight bend in the knee for several reasons related to stability and efficiency:

1. **Stability**: A slightly bent posture helps improve balance and stability. This is similar to how humans naturally stand with some degree of flexion at the knees. The bending allows the robot's center of mass to be more centrally positioned over its base, reducing the risk of tipping or falling.
2. **Energy Efficiency**: Bent knees can help reduce energy consumption during static stance phases. By maintaining this position, the mechanical systems (such as motors) require less effort to support the weight of the body without additional strain on joints that might occur if they were fully extended.
3. **Dynamic Stability During Movement**: When performing dynamic movements such as walking or running, having flexible joint angles like those found in human anatomy enables better control and adaptability. Bipedal robots often use these principles to enhance their ability to navigate various terrains and respond to external forces dynamically.
4. **Biological Inspiration**: Many robotic designs draw inspiration from biological models, including human locomotion patterns. Mimicking natural postures can lead to more effective and efficient movement strategies.
5. **Reduction of Stress on Joints and Components**: Keeping certain joints moderately flexed reduces stress on both the physical components within the robot and potential wear and tear associated with prolonged periods of full extension.

These factors collectively contribute to why many bipedal robots adopt a posture involving mild knee bends while standing or moving. However, specific design choices may vary based on the intended application, capabilities, and goals of each individual robot model.

---

$\alpha = 20$ (Formal)

---

Most bipedal robots maintain a slight bend in the knee for several reasons:

1. **Stability and Balance**: Bending the knees slightly helps to enhance stability and balance. This posture allows the robot to distribute its weight more effectively across the lower limbs, reducing the risk of tipping or imbalance.
2. **Postural Adaptation**: Human beings naturally adopt such postures due to biomechanical advantages that facilitate efficient movement and support various activities. Similarly, robotic designs often mimic human postures to improve functionality and adaptability.
3. **Flexibility and Dexterity**: A slight bend in the knees provides greater flexibility and dexterity, enabling the robot to perform a wider range of actions with ease. It facilitates movements such as walking, climbing, and manipulation tasks.
4. **Energy Efficiency**: Bent knees can contribute to energy efficiency during locomotion by allowing smoother transitions between different phases of gait (such as stance phase and swing phase).
5. **Safety Considerations**: Maintaining an upright yet flexible posture reduces stress on joints and musculature over prolonged periods, which is crucial for long−term operation without fatigue or damage.
6. **Mimicry of Natural Movements**: Many robotic systems aim to emulate natural human behaviors to better integrate them into environments where they interact closely with humans, thereby enhancing social acceptability and effectiveness.

These factors collectively explain why many bipedal robots employ a posture characterized by mild flexion at the knees while maintaining other aspects similar to those observed in human physiology.

---

## J  CASE OF CROSS-LANGUAGE GENERALIZATION

To further evaluate the cross-lingual capability of our model, we tested whether activation vectors extracted from English datasets could control formality in **German** question-answering tasks. The experimental results, presented below, demonstrate that our activation vectors effectively generalize across languages, maintaining robust control over stylistic attributes in the target language.

**Question:** Die globale Finanzkrise von 2007/2008. Wird es eine weitere geben?

$\alpha = 20$ (Informal)

> Es ist schwierig zu sagen, wann und wie das nchste groe Finanzspielzeug wird, aber man kann sicher sein, dass es immer noch Risiken gibt, die dazu fhren knnen, dass der Markt pltzlich in Schwierigkeiten gert.
> Hier sind einige Grnde, warum ein neuer Finanzschlafmtze vielleicht kommen knnte:
> 1. **Hilfsschlafmtzen**: Wenn jemand versucht, den Markt mit Hilfe von Geld aus dem Nichts machen will – sozusagen einen groen Spielball fr sich selbst – dann kann das zum nchsten groen Schlafmtzenspiel werden.
> *Beispiel:* Wie bei den Hauseinlagen, die vor vielen Jahren als super–sicher gedacht wurden, aber im Endeffekt viele Leute tiefen Schlfer aufwachen lieen, wenn sie ihre Hausgeldkarten nicht gut genug versteckt hielten oder vergaen, dass sie gar kein echtes Geld waren!
> 2. **Zu viel Geld suchen nach einem Ort, um zu spielen**:
> *Wenn alle Banken und andere Menschen, die Geld haben wollen, keine gute Idee finden, wo sie ihr Geld lassen sollen, dann knnten sie versuchen, etwas ganz Neues zu finden!*
> 3. **Menschen, die glauben, dass alles okay ist**, wenn es nicht wirklich okay ist:
> *Wie wenn du weit, dass dein Auto kaputt ist, aber du fhrst trotzdem damit, weil du denkst, dass es schon lange keinen Fehler gemacht hat.*
> 4. **Ganze Lnder, die dummspielen**:
> *Was passiert, wenn alle Leute in einer Stadt oder Land, das sehr viel Geld hat, einfach nur Spa haben wollen, ohne richtig darber nachdenken, was genau sie tun? Es wre wie wenn alle Freunde in deinem Kindergarten zusammenhelfen wrden, um ein riesiges Sandburg–Haus zu bauen, aber niemandem kommt in den Kopf, dass es Zeit ist, mal rber zur Wasserbrunnen–Schaukel zu gehen!*

$\alpha = -20$ (Formal)

> Es ist schwierig zu sagen, ob und wann genau eine neue Finanzkrise passieren wird, aber wir knnen sicherstellen, dass das Risiko immer da ist. Hier sind einige Punkte zum Thema Finanzrisiken:
> 1. **Lernende Systeme**: Nach der groen Krise von 2007/2008 haben viele Lnder und Banken versucht, ihre Systeme besser zu machen. Das bedeutet nicht, dass sie perfekt sind oder nie wieder Probleme haben werden.
> 2. **Debt–Bubble**: Eine groe Schuldenbombe kann sich bauen, wie in den Jahren vor dem Crash. Wenn Leute, Unternehmen oder Regierungen zu viel Geld leihen, ohne genug Kredite zurckzahlen zu knnen, kann das zu einem Problem fhren.
> 3. **Housing–Markt–Schwankungen**: Immobilienblscherei war ein groer Teil des Problems im Jahr 2007/2008. Wenn der Immobilienmarkt pltzlich abfllt, kann das einen Sturz starten.
> 4. **Technologiefirmen**: Heutzutage gibt es andere Sektoren, die riesige Betten voller Geld haben knnten  z.B. Tech–Firmen mit hohen Bewertungen. Ein Schlag fr diese Firmen knnte alles verndern.
> 5. **Klimawandel**: Die Vernderte Weltwirtschaft durch den Klimawandel kann auch zu neuen Art von Krisen fhren, die uns berraschen knnten.
> 6. **Politik und Rechtswesen**: Politische Unruhe und Schwierigkeiten bei der Steuerung von Marktsystemen knnen auch dazu fhren, dass Dinge aus dem Ruder laufen.
> Um solche Krisen zu verhindern oder wenigstens zu dmpfen, mssen Lnder und Institutionen stndig aufpassen und sich anpassen. Es hilft auch, wenn jeder seine eigene finanzielle Situation gut kontrolliert und keine zu vielen Risiken nimmt. Insgesamt ist es wichtig, dass alle Beteiligten vorsichtig bleiben und immer nach besseren Mglichkeiten suchen, um Risiken zu minimieren.

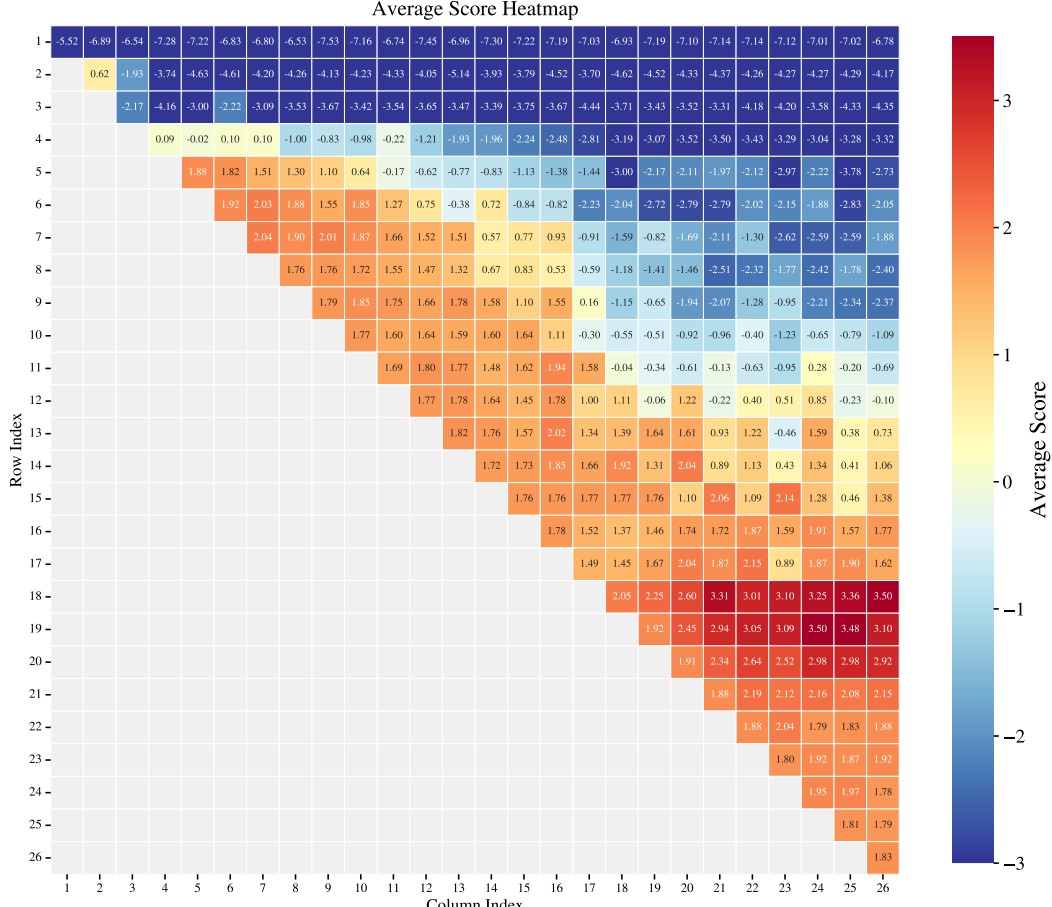

Figure 6: **Heatmap of average style control scores across layer blocks.** The y-axis and x-axis represent the start and end indices for the intervention block, respectively. Scores are averaged over multiple runs.

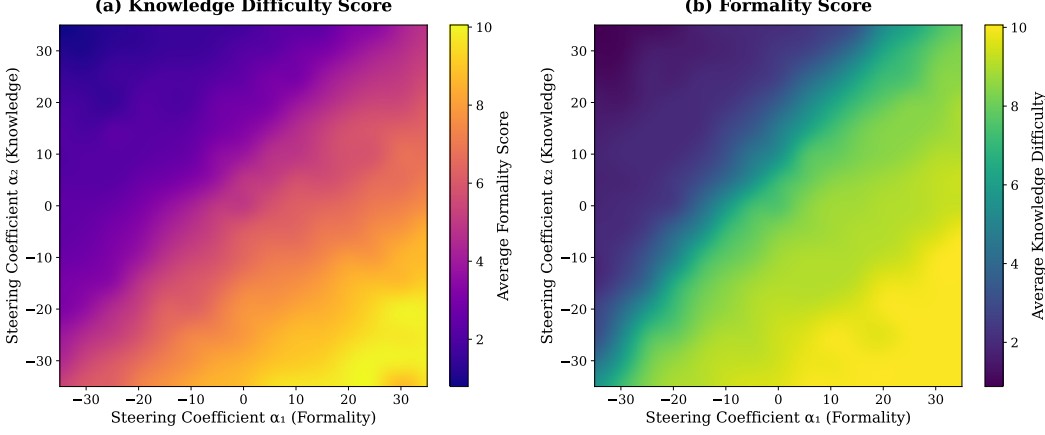

Figure 7: **Two-Dimensional Stylistic Control.** Model output scores for (a) Knowledge Difficulty and (b) Formality under simultaneous steering.

