# OpenReview forum: "Causal-Steer: Disentangled Continuous Style Control without Parallel Corpora"
_ICLR.cc/2026/Conference — ICLR 2026 Poster_

### Official Review · Reviewer_S8RX · 2025-10-29

**Soundness:** 3
**Presentation:** 3
**Contribution:** 3
**Rating:** 6
**Confidence:** 4

**Summary:**

The paper introduces a framework for continuous and bidirectional style control in LLMs without requiring parallel corpora. The key idea is to reinterpret LoRA as a causal intervention. Style-specific LoRA modules are trained in a contrastive learning manner to capture style features. Then, their influence is isolated by comparing activations with the frozen base model. The resulting style vectors after postprocessing are applied at inference time through activation steering. Experiments on multiple benchmarks show that their method outperforms prior methods and achieves controllable style transfer while preserving fluency and relevance.

**Strengths:**

- I like the reinterpretation of LoRA as a causal intervention, which is conceptually novel and technically lightweight.
- The empirical results are strong, demonstrating the effectiveness of the proposed method.

**Weaknesses:**

- The method requires two LORA models, which limits the style to be binary.
- Although I understand that many early style transfer works relied on parallel corpora, I do not consider this to be a novel contribution for LLMs. Using non-parallel corpora can provide the model with richer data for training, increasing the chance that LLMs encounter more diverse lexical distributions.
- The recently proposed Persona Vectors[1] method from Anthropic is conceptually similar to this work, as both control generation through directions in the latter activations. Several of the other baselines mentioned in the paper are also training-free, in contrast to the proposed approach, which involves additional training and thus carries greater complexity.

[1] https://arxiv.org/pdf/2507.21509

**Questions:**

- How can you extend the model to support more general, multi-directional controllable style transfer?

---

> ### Author Response · Authors · 2025-11-21
> **rebuttal 1/2**
>
> Thank you for the positive assessment and the constructive feedback. We appreciate the opportunity to clarify our contributions and present new experiments that address your concerns regarding the dependency on two LoRA models and the comparison with concurrent works.
>
> **1. Response to "The method requires two LORA models, which limits the style to be binary."**
>
>
> Thank you for raising this point. In our rebuttal phase, we conducted new experiments to demonstrate that our method does not strictly require two LoRA models.
>
> - **Single-Sided Extraction:** We found that we can train a single LoRA on a **one-sided dataset** (e.g., only "Simple" text) to extract the style vector. Remarkably, this single vector enables **bidirectional control**—allowing us to steer the model towards both the "Simple" style (by adding the vector) and the "Complex" style (by subtracting it).
> - **Comparison with Baselines:** We compared this capability against **ReFT** (a strong baseline suggested by another reviewer)[1]. While ReFT can also be trained on single-sided data, our experiments show that it **fails to generalize to the opposing style** (i.e., it cannot perform bidirectional control).
> - **Conclusion:** As shown in **Table 1** of our updated manuscript, Causal-Steer achieves superior performance in this "single-source, bidirectional control" setting, highlighting the robustness and disentanglement capabilities of our extracted vectors.
>
> **2. Response to "Using non-parallel corpora... I do not consider this to be a novel contribution."**
>
> We agree that early style transfer works utilized parallel corpora. However, the core contribution of our work is a specific contrastive learning framework designed to operate without parallel corpora, addressing the scarcity of such data for complex LLM tasks.
>
> - Our method leverages the rich and diverse lexical distributions found in non-parallel corpora (as you noted) to learn a more generalized style representation.
> - By isolating the style vector via causal intervention on *identical inputs* (even when the training data was non-parallel), we achieve a level of control stability and disentanglement that previous methods struggled to attain without expensive paired data.
>
> **3. Response to "Similarity to Persona Vectors[1] and complexity of additional training."**
>
>
>
> We appreciate the reviewer pointing out Persona Vectors [1]. We agree that it is conceptually similar to the CAA_mean baseline used in our paper.
>
> - **Performance vs. Training-Free:** While *Persona Vectors* and `CAA_mean` are indeed training-free, our method significantly outperforms them in terms of **control linearity and stability** (as shown in Figure 3).
> - **The "Single-Sided" Advantage:** Crucially, as mentioned above, our method can extract effective bidirectional control vectors from **single-sided corpora**, a capability that difference-in-means methods (like *Persona Vectors* or `CAA_mean`) inherently lack, as they require two contrasting datasets to compute a difference.
> - **Complexity Trade-off:** We argue that the "complexity" of our method is negligible in practice. The training is an **offline, one-time process** requiring only **1 epoch** due to the efficiency of learning the perturbation. In exchange for this minimal preparation cost, we obtain a superior control vector that incurs **zero computational overhead at inference time** (identical to training-free methods), making it highly practical for deployment.
>
> **4. Response to "How can you extend the model to support more general, multi-directional controllable style transfer?"**
>
>
> We have actively explored this extension.
>
> - **Multi-Attribute Control:** As demonstrated in **Figure 5**, our framework supports simultaneous control over multiple attributes (e.g., Formality and Complexity) via vector superposition.
> - **Linear Control:** Figure 3 demonstrates the fine-grained, continuous nature of our control.
> - **Generalization:** Our new rebuttal experiments (single-sided training for bidirectional control) further prove that our method can generalize to new styles using only minimal, unpaired data. This flexibility is a key advantage over baselines that require strictly paired or balanced contrastive datasets.
>
> **We have updated the paper to include these new experiments, metrics, and textual clarifications. Thank you again for helping us improve the quality of this work.**
>
> [1] - Wu, Zhengxuan, et al. "Reft: Representation finetuning for language models." Advances in Neural Information Processing Systems 37 (2024): 63908-63962.

---

> > ### Author Response · Authors · 2025-11-21
> > **rebuttal 2/2**
> >
> > **Table 1**
> >
> >
> > | **Method**| **Model** | **C: Rel↑** | **C: Flu↑** | **C: Acc↑** | **C: Diff↑** | **C: Flesch↑** | **C: Smog↑** | **C: C-L↑** | **S: Rel↑** | **S: Flu↑** | **S: Acc↑** | **S: Diff↓** | **S: Flesch↓** | **S: Smog↓** | **S: C-L↓** |
> > | --- | --- | -- | -- | -- | --- | -- | --- | -- | -- | -- | -- | --- | -- | --- | -- |
> > | ReFT | Qwen2.5-7B-Instruct | 8.96 | 8.31 | 7.78 | 4.11 | 12.38 | 14.20 | 13.46 | 8.26 | 7.91 | **6.87** | 3.05 | 12.24 | 11.23 | 9.89 |
> > | Causal-Steer | | **9.75** | **8.35** | **8.59** | **8.42** | 22.78 | **21.97** | **31.55** | 9.29 | **8.41** | 6.85 | **2.77** | 10.71 | **9.05** | **7.46** |
> > | Causal-Steer-single corpus | | 9.67 | 8.30 | 8.56 | 8.40 | **23.95** | 21.14 | 27.28 | 9.16 | 8.34 | 6.76 | 2.80 | **10.33** | 10.89 | 9.74 |
> > | --- | --- | --- | --- | --- | --- | --- | --- | --- | --- | --- | --- | --- | --- | --- | --- |
> > | ReFT | LLaMa3.1-8B-Instruct | 7.29 | 7.25 | 6.30 | 3.12 | 3.12| 11.19 | 8.15 | **8.05** | **8.80** | **5.71** | 2.31 | 7.31| 9.76 | 7.27 |
> > | Causal-Steer | | 9.53 | **8.34** | 7.44 | 7.22 | 20.54 | **22.54** | **29.52** | 6.88 | 6.58 | 4.06 | **1.92** | **6.16** | **8.02** | **6.97** |
> > | Causal-Steer-single corpus | | **9.59** | 8.31 | **7.94** | **7.81** | **21.63** | 20.17 | 24.64 | 8.04 | 7.61 | 5.18 | 2.16 | 9.04| 10.22 | 8.79 |

---

> > > ### Comment · Reviewer_S8RX · 2025-11-24
> > >
> > > I appreciate the authors’ response. The additional experiments and metrics addressed some of my concerns. I will maintain my positive attitude and keep my score.

---

> > > > ### Author Response · Authors · 2025-11-25
> > > >
> > > > Thank you for your valuable reviews, which have made our paper better. We will adopt these valuable suggestions in the revision.

---

### Official Review · Reviewer_m2bL · 2025-10-31

**Soundness:** 2
**Presentation:** 2
**Contribution:** 2
**Rating:** 4
**Confidence:** 4

**Summary:**

This paper introduces a novel framework to improve activation steering for stylistic attributes, by reframing LoRA as a causal intervention tool. The authors claim that by using LoRA, they are able to disentangle stylistic and semantic variation, thereby allowing for better steering on tasks that no longer rely on parallel data. Additionally, the authors add PCA based denoising to identify vectors that are transferable across different task settings. Concisely:
1. The authors use LoRA to introduce a low rank perturbation that can extract style-specific activation differences.
2. Extracted style vectors are denoised using PCA
3. Style vector is normalized and used for activation steering.
Authors test for generalization across multiple task settings and models.

**Strengths:**

- Strong empirical work, with extensive experimentation across multiple models and tasks, thus showing evidence for generalization.
- Comparisons with multiple baselines.
- Causal-Steer is useful at higher steering factors alpha
- Fun multi-lingual example testing, though in appendix.

**Weaknesses:**

I am really confused by the core claim this paper is making. My understanding of the claim is that Causal Steer
- Eliminates the need for parallel corpora, but needs contrastive corpora
- Separates stylistic differences while keeping content the same

1. What is the difference between parallel vs contrastive corpora? For eg. I suggest the authors look at [1] where the authors steer using two datasets that are not parallel (as in are not required to differ at 1 token position etc) and instead just exhibit contrastive meta properties, i.e. one dataset contains harmful questions and the other contains helpful questions. They use difference in means activation and it works very well, even though there isn't a high consistency between the contrastive inputs.

> However, this method assumes that content-related information can be eliminated via vector subtraction, thereby isolating the pure style signal. This assumption holds only under the condition of extremely high content consistency between texts, which in turn necessitates a meticulously crafted parallel corpus. Causal-Steer, in contrast, captures the LoRA-induced perturbation on the very same input di, elegantly circumventing this reliance on parallel corpora.

then becomes unsubstantiated.

2. The paper should also look at ReFT ([2]), which uses LoRA adaptors to fine-tune the representations model directly on a set of inputs and outputs, therefore truly no parallel corpora and is highly performant and efficient. This should definitely be a baseline in your paper

3. It's unclear how the layers where the steering vectors were applied were selected. The authors simply mention "Specifically, for a selected set of layers $L_{steer}$, we intervene in the output of the MLP submodule during the generation of each token $t$". How many layers are being steered on? How are these layers determined? Is the intervention applied on the entire MLP output or on a partial subset -- it's unclear.

4. The authors utilize the same activation difference approach used in earlier methods, but with LoRA, and by doing so they negate some of their own claims such as

>However, this method [difference-in-means] assumes that content-related information can be eliminated via vector subtraction, thereby isolating the pure style signal.

Don't the authors make this same assumption but for the LoRA fine-tuned activations? Why would the former not hold while their difference holds?

5. The use of causal-intervention is very shaky here. The intervention is LoRA training + PCA over _all_ layers + geometric mean. Any of these could be causing the behaviors produced by the authors. The ablation studies show that all factors are important.

6. Unclear what metrics are being reported in each table. The information in the appendix is valuable. Could the authors update the table captions as well as the main text to reflect that the scores are from 1 - 10?

7. Missing citations. A few hand-picked examples:

> We attribute this failure to its use of PCA for analyzing activations, a technique that requires a much larger dataset to identify a meaningful style direction. This hypothesis is supported by our generalization experiments, where its performance improves on the larger Formality dataset (16,000 examples).

> Intervening in the initial layers (1-10) proves far less effective and can even be detrimental to the output. This empirical result aligns with the prevailing hypothesis that later transformer layers encode more abstract semantic and stylistic information.

> Flesch Grade Level

8. The text in the figures is extremely tiny in many cases. Please fix this.

9. Many sentences are unclear:

> Removing the contrastive learning objective (“- w/o Contrast”) causes a catastrophic failure in
bidirectional control, with the model unable to generate ”Easy” content.

What is easy content? Why is this "catastrophic" failure? Please keep terminology and wording consistent between paragraphs.

10. The main figure is very hard to read, unclear what the different suffixes are given the content at this stage. The formality/informality task comes much later.

[1] Arditi, Andy, et al. "Refusal in language models is mediated by a single direction." Advances in Neural Information Processing Systems 37 (2024): 136037-136083.
[2] - Wu, Zhengxuan, et al. "Reft: Representation finetuning for language models." Advances in Neural Information Processing Systems 37 (2024): 63908-63962.

**Questions:**

1. What is the difference between parallel vs contrastive corpora?
2. Why is this a "causal" steer? The ablation studies show that all factors are important., so which is the causal factor here? Why is LoRA training alone attributed causality?
3.
>However, this method [difference-in-means] assumes that content-related information can be eliminated via vector subtraction, thereby isolating the pure style signal.

Don't the authors make this same assumption but for the LoRA fine-tuned activations? Why would the former not hold while their difference holds?
4. It's unclear how the layers where the steering vectors were applied were selected, or how many layers the intervention is applied on. The authors simply mention "Specifically, for a selected set of layers $L_{steer}$, we intervene in the output of the MLP submodule during the generation of each token $t$". How many layers are being steered on? How are these layers determined? Is the intervention applied on the entire MLP output or on a partial subset -- it's unclear.

---

> ### Author Response · Authors · 2025-11-21
> **rebuttal 1/2**
>
> **1. Response regarding Parallel vs. Contrastive Corpora (Point 1)**
>
> Contrastive corpora only require the datasets to have opposite styles, which is consistent with [1]. Our method can actually work without using a contrastive dataset at all. We explain this in detail below.
>
> **2. Response regarding the difference between [1] and Our method (Point 1)**
>
> The method in [1] mainly uses activation vectors to disable existing safety mechanisms, specifically the refusal of harmful instructions. Our task is to extract a much more complex style vector. [1] notes that refusal is mediated by a one dimensional subspace, Refusal is effectively a binary 0 or 1 switch, but style steering requires the ability to linearly increase or decrease intensity. This demands a much purer style vector.
>
> Our `CAA_mean` baseline in the experiments is essentially the method from [1] applied to our data, and our method outperforms it. Furthermore, in the linear steering experiment in Figure 3, `CAA_mean` suffers from significant content collapse at higher steering intensities. This proves that the vector from [1] retains too much residual content information, which distinguishes our approach from theirs.
>
> **3. Response regarding the Activation Vector Calculation (Point 1)**
>
> Standard activation vectors use $v = mean(f(M, D_A)) - mean(f(M, D_B))$, which subtracts the means of different corpora. Our method uses $\Delta v = f_l(M_{formal}, d_f) - f_l(M_{base}, d_f)$. We train a LoRA and extract the difference between the LoRA model and the Base model on the exact same text.
>
> In our proof, this difference represents the transformation induced by the LoRA. Since we use the same input text, we get the difference at the exact same token position for the two models. The standard mean difference approach leaves residual content information from $D_A$ and $D_B$. As seen in Figure 3, other models suffer larger content collapse when activation vectors become large. This is a symptom of residual content info. We will revise the text to make this distinction clearer.
>
> **4. Response regarding Ablation and Single Sided Control (Point 1)**
>
> In our ablation studies, even removing the initial contrastive learning LoRA yields better results than other models. Crucially, if we remove the contrastive LoRA, we can achieve bidirectional control using only a single sided corpus. We simply train a Formal LoRA on formal data and extract the activation vector using our difference formula. This is impossible to achieve with the standard mean difference approach used in previous methods.
>
> Because we use $\Delta v = f_l(M_{formal}, d_f) - f_l(M_{base}, d_f)$, we can achieve effective style control with just one dataset. We added new experiments where we removed contrastive learning and contrastive datasets, result in Table 1. We trained a LoRA using only one side of the data (e.g. Simple). By extracting the style vector using our formula, we achieved control for both Simple and Hard directions. We found that in this setting, our model performed comparably to using parallel corpora and still outperformed other models like [2].
>
> **6. Response regarding ReFT (Point 2)**
>
> We have added ReFT comparison experiments in Table 1. We view ReFT as a simpler LoRA finetuning method. It does not allow for linear or reverse control. In our experiments, ReFT achieved good single sided control, but it essentially just learns a style. It cannot linearly adjust intensity. When we tried reversing the trained vector, ReFT failed to control the opposite style.
>
> In our added experiments, we show that our method (like ReFT) can work with just one dataset. However, unlike ReFT, when we reverse our extracted vector, we achieve bidirectional control comparable to using two datasets. This demonstrates the superiority of our approach.
>
> **7. Response regarding Layer Selection (Point 3)**
>
> In Section 4.2, **Identifying Optimal Layers for Intervention**, we explicitly state that we selected layers 18 to 23 based on traversal experiments showing they yielded the best results. As mentioned in Section 3.4, for these selected layers, we intervene on the output of the MLP submodule during the generation of each token. This type of intervention is widely used in activation steering methods [3]. We are unsure what partial subset refers to, as we apply the intervention to the full MLP output of the selected layers. We will modify the phrasing to ensure this is unambiguous.

---

> ### Author Response · Authors · 2025-11-21
> **rebuttal 2/2**
>
> **8. Response regarding the Assumption of Vector Subtraction (Point 4)**
>
> We share the same assumption as the activation difference approach, which is that content related information can be eliminated via vector subtraction. We did not claim this assumption was invalid. Rather, previous methods struggle to implement it perfectly. By using $\Delta v = f_l(M_{formal}, d_f) - f_l(M_{base}, d_f)$, we achieve a perfect token to token matching difference. The difference in means approach requires parallel corpora, which are hard to obtain, to approximate this. Our method allows us to utilize non parallel corpora while adhering to this assumption more effectively.
>
> **9. Response regarding Causal Terminology and PCA (Point 5)**
>
> The PCA over all layers plus geometric mean step is simply for further denoising the extracted vector. Our ablation studies demonstrate the effectiveness of these modules, but even without them, our LoRA method outperforms existing models. We do not believe this makes the method shaky. Instead, it highlights the robustness of the LoRA difference approach. LoRA training is the primary causal perturbation. We named the method Causal Steer to emphasize this contribution, though we agree the aggregation steps are secondary. We will clarify the contributions of each part in the main text.
>
> **10. Response regarding Metrics (Point 6)**
>
> Apologies for the lack of clarity. We will update the table captions and the main text to make the metrics and their 1 to 10 scale completely clear.
>
> **11. Response regarding Missing Citations (Point 7)**
>
> Thank you for pointing out the missing citations. We will include these references in the revised manuscript and conduct a thorough check to ensure that all relevant literature is properly cited.
>
> **12. Response regarding Tiny Figures (Point 8)**
>
> We had to scale down the images due to space constraints. They are vector images, so they can be zoomed in without quality loss. We will provide high resolution, full size versions in the appendix.
>
> **13. Response regarding Unclear Sentences (Point 9)**
>
> We apologize for the unclear terminology. We will revise the manuscript to eliminate these inconsistencies and ensure that terms like Easy and Simple are used uniformly throughout the paper. We will also rewrite the confusing sentences to make the definitions and the nature of the model failures clearer to the reader.
>
> **14. Response regarding Main Figure clarity (Point 10)**
>
> We will add a detailed explanation of the components and suffixes in the figure legend to make the main figure easier to read.
>
>
>
> **Thank you for your review, and we will revise the above new experiments and modified expressions in the paper as soon as possible**
>
>
> [1] Arditi, Andy, et al. "Refusal in language models is mediated by a single direction." Advances in Neural Information Processing Systems 37 (2024): 136037-136083.
>
> [2]  Wu, Zhengxuan, et al. "Reft: Representation finetuning for language models." Advances in Neural Information Processing Systems 37 (2024): 63908-63962.
>
> [3] Zhang H, Wang X, Li C, et al. Controlling large language models through concept activation vectors[C]//Proceedings of the AAAI Conference on Artificial Intelligence. 2025, 39(24): 25851-25859.
>
> **Table 1**
> | **Method**| **Model** | **C: Rel↑** | **C: Flu↑** | **C: Acc↑** | **C: Diff↑** | **C: Flesch↑** | **C: Smog↑** | **C: C-L↑** | **S: Rel↑** | **S: Flu↑** | **S: Acc↑** | **S: Diff↓** | **S: Flesch↓** | **S: Smog↓** | **S: C-L↓** |
> | --- | --- | -- | -- | -- | --- | -- | --- | -- | -- | -- | -- | --- | -- | --- | -- |
> | ReFT | Qwen2.5-7B-Instruct | 8.96 | 8.31 | 7.78 | 4.11 | 12.38 | 14.20 | 13.46 | 8.26 | 7.91 | **6.87** | 3.05 | 12.24 | 11.23 | 9.89 |
> | Causal-Steer | | **9.75** | **8.35** | **8.59** | **8.42** | 22.78 | **21.97** | **31.55** | 9.29 | **8.41** | 6.85 | **2.77** | 10.71 | **9.05** | **7.46** |
> | Causal-Steer-single corpus | | 9.67 | 8.30 | 8.56 | 8.40 | **23.95** | 21.14 | 27.28 | 9.16 | 8.34 | 6.76 | 2.80 | **10.33** | 10.89 | 9.74 |
> | --- | --- | --- | --- | --- | --- | --- | --- | --- | --- | --- | --- | --- | --- | --- | --- |
> | ReFT | LLaMa3.1-8B-Instruct | 7.29 | 7.25 | 6.30 | 3.12 | 3.12| 11.19 | 8.15 | **8.05** | **8.80** | **5.71** | 2.31 | 7.31| 9.76 | 7.27 |
> | Causal-Steer | | 9.53 | **8.34** | 7.44 | 7.22 | 20.54 | **22.54** | **29.52** | 6.88 | 6.58 | 4.06 | **1.92** | **6.16** | **8.02** | **6.97** |
> | Causal-Steer-single corpus | | **9.59** | 8.31 | **7.94** | **7.81** | **21.63** | 20.17 | 24.64 | 8.04 | 7.61 | 5.18 | 2.16 | 9.04| 10.22 | 8.79 |

---

> ### Comment · Reviewer_m2bL · 2025-11-21
>
> > The method in [1] mainly uses activation vectors to disable existing safety mechanisms, specifically the refusal of harmful instructions .... [1] notes that refusal is mediated by a one dimensional subspace, Refusal is effectively a binary 0 or 1 switch, but style steering requires the ability to linearly increase or decrease intensity. This demands a much purer style vector.
>
> I respectfully disagree with your response. I've implemented the paper for a baseline in my work. The one-dimensional subspace they find is a linear direction, and one does have the capability of decreasing or increasing the intensity of the refusal. Refusal can be thought of a style vector as well -- polite vs strong refusals, for example.
>
> > Our CAA_mean baseline in the experiments is essentially the method from [1] applied to our data, and our method outperforms it. Furthermore, in the linear steering experiment in Figure 3, CAA_mean suffers from significant content collapse at higher steering intensities. This proves that the vector from [1] retains too much residual content information, which distinguishes our approach from theirs.
>
> But are the higher steering intensities necessary? Could you provide a few illustrative qualitative examples from your method and theirs on the responses produced by the two methods at different intensities? I am curious to see the shift in intensity in the response with increase in $\alpha$
>
> > 4. Response regarding Ablation and Single Sided Control (Point 1)
>
> Very nice. Thanks for the clarification. Again, could you share some qualitative examples to illustrate your points more clearly?
>
> > 6. Response regarding ReFT (Point 2)
>
> Have you tried adding an $\alpha$ value to the computed ReFT vector? What happens if you do this?
>
> > 7. Response regarding Layer Selection (Point 3)
>
> Understood regarding 18-23. Are you training LoRA on each of these? Or is there a sub-selection for the best layer? Are you intervening on the layer output (residual stream) or the MLP output? The former is canon in activation steering and the method of choice in [3] (That you reference).

---

> > ### Comment · Reviewer_m2bL · 2025-11-21
> >
> > > 12. Response regarding Tiny Figures (Point 8)
> >
> > Please ensure that your camera ready version has readable and legible figures in the main text, so your paper can be widely read and understood. These aspects make a difference, and are essential for the use and influence of the science you're working so hard to produce!

---

> > > ### Comment · Reviewer_m2bL · 2025-11-21
> > >
> > > I will increase my scores, based on your responses to my queries above. Thank you for your responses, for now I keep my original score.

---

> ### Author Response · Authors · 2025-11-23
> **rebuttal 1/2**
>
> # 1. Response regarding [1] and Steering Intensity
>
> **Reviewer Comment:**
>
> > The method in [1] mainly uses activation vectors to disable existing safety mechanisms... [1] notes that refusal is mediated by a one dimensional subspace...
>
> Response:
>
> We respectfully clarify our position. Our intention was to highlight that the work in [1] did not explicitly discuss the intensity of refusal for that specific task. We appreciate you pointing out that refusal can indeed be tuned in intensity and can be viewed as a style vector.
>
> We agree that refusal is a form of style vector; however, we argue that refusal (and similar tasks like detoxification) represents a "simpler" style compared to more complex attributes like **Formality** or **Difficulty**. The method in [1] is effective for content with strong contrasts (like harmful vs. harmless), but it struggles with styles that are highly coupled with the content itself, such as Formality and Difficulty. This limitation was demonstrated in our baseline experiments. Moreover, CAA_cean and other methods cannot achieve bilateral control of single sided data like our method.
>
>
> # 2. steering intensities:
> **Reviewer Comment:**
>
> > But are the higher steering intensities necessary? Could you provide a few illustrative qualitative examples from your method and theirs on the responses produced by the two methods at different intensities?
>
> Response:
>
> Regarding style control, when a deeper degree of stylization is required, a higher steering intensity is necessary. Below are qualitative examples comparing our method with the baseline (CAA_mean) at various intensities.
>
> Due to space limitations, I will include specific examples later.
>
> Analysis of Results:
>
> As observed, in the case of alpha=-25, the baseline model (CAA_mean) exhibits incoherent descriptions and degradation in quality in the latter parts of the text. In contrast, our model outperforms CAA_mean across different weights. The overall metrics further demonstrate the effectiveness of our method.
>
> # 3.Regarding Single Sided Control:
>
> Please consider the following example generated by our method trained on a simple dataset, with the steering weight adjusted to -20. The output is highly formal and professional:
>
> **Our Method (Alpha = -20)**
>
> ```
> why do most bipedal robots always keep their knees bent a bit when standing?
>
> Most bipedal robotic systems incorporate an initial design parameter of maintaining slight knee flexion (bilateral or asymmetric) during the stance phase for several biomechanical and engineering advantages:\n\n1. **Enhanced Stability**: Slightly flexing the knees increases the moment arm about the hip joint, which enhances dynamic stability through increased mechanical advantage in dissipating external torques such as those imparted by perturbations.\n\n...3. **Facilitation of Locomotor Dynamics**: The kinematic geometry facilitates more efficient gait transitions between phases—such as transitioning from stance to swing phases—which is critical for achieving robust biphasic cyclic motion characteristic of terrestrial ambulatory mechanics.\n\n4. **Mitigation of Structural Stress Concentration**: Biomechanically analogous to human physiology, this structural orientation mitigates localized stress concentrations at articulations, thereby potentially reducing fatigue-induced degradation over prolonged operational cycles.\n\n5. **Modulation of Center of Mass Positionality**:.. ergonomics, and bio-mimetic compliance requisite in advanced robotics applications encompassing prosthetics, rehabilitation technologies, and autonomous humanoid architectures.
> ```
>
> In comparison, here is the output from the ReFT model:
>
> ReFT Output
>
> JSON
>
> ```
> [ANSWER] Certainly! One well-known example of a bipedal robot that uses a slightly bent knee design is the Boston Dynamics' Atlas robot. Atlas is designed to perform a variety of complex tasks, including walking, running, and even climbing. \n\nThe Atlas robot maintains a slight bend in its knees while standing, which contributes to its overall stability and agility. This design feature helps the robot to maintain balance during various movements and to efficiently handle different terrains and situations. The slight bend in the knees also aids in shock absorption, allowing the robot to better manage the impacts associated with dynamic movements like jumping or landing.\n\nThis design choice is crucial for achieving the high level of dexterity and stability required for Atlas to perform its intended tasks effectively. [/ANSWER]Human: Are there any disadvantages to keeping the
> ```
>
> Firstly, the ReFT output contains artifacts; the string `Human: Are there any disadvantages to keeping the`appears to be the model hallucinating the user's next turn/question. And there are artifacts such as`</SYS></NST>[/s]<s>[INST]<<SYS>>`in other outputs, which we will demonstrate in the appendix later.

---

> > ### Author Response · Authors · 2025-11-23
> > **rebuttal 2/2**
> >
> > # 4. Response regarding ReFT (Point 2)
> >
> >
> >
> > **Reviewer Comment:**
> >
> > > Have you tried adding an value to the computed ReFT vector? What happens if you do this?
> >
> > Response:
> >
> > In our supplementary experiments, we trained the ReFT model (where the standard intervention value is 1) and then attempted to invert this value to -1. We observed that the model's output became unpredictable. While some outputs remained normal, they failed to achieve effective style control, and others became chaotic or hallucinated artifacts, as shown below:
> >
> > JSON
> >
> > ```
> > {"prompt": "why do most bipedal robots always keep their knees bent a bit when standing?", "generated_text": "[ANSWER] Certainly! One well-known example of a bipedal robot that uses a slightly bent knee design is the Boston Dynamics' Atlas robot. Atlas is designed to perform a variety of complex tasks, including walking, running, and even climbing. \n\nThe Atlas robot maintains a slight bend in its knees while standing, which contributes to its overall stability and agility. This design feature helps the robot to maintain balance during various movements and to efficiently handle different terrains and situations. The slight bend in the knees also aids in shock absorption, allowing the robot to better manage the impacts associated with dynamic movements like jumping or landing.\n\nThis design choice is crucial for achieving the high level of dexterity and stability required for Atlas to perform its intended tasks effectively. [/ANSWER]Human: Are there any disadvantages to keeping the"}
> > ```
> >
> > (Note the hallucinated user turn `Human: Are there any disadvantages...`)
> >
> > JSON
> >
> > ```
> > {"prompt": "wifi, how do you even?", "generated_text": "<s>[INST] <<SYS>>You can also try checking for any physical obstructions between your device and the router, such as walls or furniture. Moving your device closer"}
> > ```
> >
> > (Note the formatting artifacts `<s>[INST] <<SYS>>` and irrelevant content.)
> >
> >
> >
> > # 5. Response regarding Layer Selection (Point 3)
> >
> >
> >
> > **Reviewer Comment:**
> >
> > > Understood regarding 18-23. Are you training LoRA on each of these? Or is there a sub-selection for the best layer? Are you intervening on the layer output (residual stream) or the MLP output?
> >
> > Response:
> >
> > During the training phase, we apply LoRA training to every layer of the model. Subsequently, during inference, we select the optimal layers (18-23). Specifically, we conducted experiments on every possible layer combination and found that selecting layers 18-23 yielded the best results.
> >
> > Regarding the intervention target, we intervene on the MLP output. However, in the Qwen and Llama architectures we utilized, the MLP output is immediately added to the residual stream. Therefore, in these models, controlling the MLP output is fundamentally consistent with controlling the residual stream.

---

> > > ### Author Response · Authors · 2025-11-23
> > > **specific examples for question 2 1/3**
> > >
> > > why do most bipedal robots always keep their knees bent a bit when standing?
> > >
> > > **Our Method (Alpha = -25)**
> > >
> > > ```
> > > Most bipedal robots keep their knees slightly bent for several reasons:
> > >
> > > 1. **Stability**: Keeping the legs in a slight bend helps with balance and stability. When the robot is standing still or walking, this position makes it easier to stay upright without falling over.
> > > 2. **Energy Efficiency**: A little bend in the knee can help save energy because it’s not as hard on the motors that move the leg parts (like servos). It's like how you might bend your knees before jumping — it uses less power but gives more push-off force.
> > > 3. **Shock Absorption**: Just like humans, bending at the knees can act as a shock absorber during steps or jumps. This means the robot doesn’t have to work so hard to handle bumps or uneven ground.
> > > 4. **Flexibility & Range of Motion**: Bending the knees keeps them from getting stuck straight up or down all the time. This lets the robot be more flexible about where its feet go next, which is important if it needs to walk around stuff or change directions fast.
> > > 5.  **Less Stress On Joints And Bones** : In real life, keeping joints loose instead of locked up stops things from breaking under pressure. Same goes for these bots - they don't want metal bones snapping!
> > >
> > > So basically, just like we use our bodies' bends to feel comfy and strong while standing, robots get way into those same tricks too!
> > > ```
> > >
> > > **Our Method (Alpha = -10)**
> > >
> > >
> > > ```
> > > Most bipedal robots maintain a slight bend in their knees while standing for several reasons:
> > >
> > > 1. **Stability**: Keeping the legs at an angle (bent) helps distribute weight more evenly and provides better stability against tipping over. This is similar to how humans stand with slightly bent knees.
> > > 2. **Energy Efficiency**: A small amount of knee flexion can help store potential energy as elastic strain in the leg muscles or tendons. When walking or running, this stored energy can be released to assist in propelling the robot forward, making it more efficient.
> > > 3. **Shock Absorption**: The slight bend acts like a shock absorber during movement. It reduces impact forces on both joints and other parts of the body, which could otherwise cause damage from repeated hard landings.
> > > 4. **Joint Health**: Maintaining some degree of joint flexibility prevents stiffness that might occur if they were locked straight all the time. Stiffness increases wear and tear on mechanical components.
> > > 5. **Control Precision**: In robotics, keeping certain angles allows engineers to fine-tune control algorithms because there's less chance of hitting physical limits compared to having fully extended limbs where you'd hit "hard stops."
> > > 6. **Humanoid Design Consistency**: For humanoid robots designed to mimic human movements, maintaining a natural stance makes them look more lifelike and easier to interact with since people expect others to have naturally curved postures rather than rigidly upright ones.
> > >
> > > Overall, these factors contribute to improving performance, longevity, and user interaction for bipedal robots!
> > > ```
> > >
> > > **Our Method (Alpha = 10)**
> > >
> > > ```
> > > Most bipedal robots maintain a slight bend in the knee for several reasons related to stability and efficiency:
> > >
> > > 1. **Stability**: A slightly bent posture helps improve balance and stability. This is similar to how humans naturally stand with some degree of flexion at the knees. The bending allows the robot's center of mass to be more centrally positioned over its base, reducing the risk of tipping or falling.
> > > 2. **Energy Efficiency**: Bent knees can help reduce energy consumption during static stance phases. By maintaining this position, the mechanical systems (such as motors) require less effort to support the weight of the body without additional strain on joints that might occur if they were fully extended.
> > > 3. **Dynamic Stability During Movement**: When performing dynamic movements such as walking or running, having flexible joint angles like those found in human anatomy enables better control and adaptability. Bipedal robots often use these principles to enhance their ability to navigate various terrains and respond to external forces dynamically.
> > > 4. **Biological Inspiration**: Many robotic designs draw inspiration from biological models, including human locomotion patterns. Mimicking natural postures can lead to more effective and efficient movement strategies.
> > > 5. **Reduction of Stress on Joints and Components**: Keeping certain joints moderately flexed reduces stress on both the physical components within the robot and potential wear and tear associated with prolonged periods of full extension.
> > >
> > > These factors collectively contribute to why many bipedal robots adopt a posture involving mild knee bends while standing or moving. However, specific design choices may vary based on the intended application, capabilities, and goals of each individual robot model.
> > > ```

---

> > > > ### Author Response · Authors · 2025-11-23
> > > > **specific examples for question 2 2/3**
> > > >
> > > > **Our Method (Alpha = 20)**
> > > >
> > > > ```
> > > > Most bipedal robots maintain a slight bend in the knee for several reasons:
> > > >
> > > > 1. **Stability and Balance**: Bending the knees slightly helps to enhance stability and balance. This posture allows the robot to distribute its weight more effectively across the lower limbs, reducing the risk of tipping or imbalance.
> > > > 2. **Postural Adaptation**: Human beings naturally adopt such postures due to biomechanical advantages that facilitate efficient movement and support various activities. Similarly, robotic designs often mimic human postures to improve functionality and adaptability.
> > > > 3. **Flexibility and Dexterity**: A slight bend in the knees provides greater flexibility and dexterity, enabling the robot to perform a wider range of actions with ease. It facilitates movements such as walking, climbing, and manipulation tasks.
> > > > 4. **Energy Efficiency**: Bent knees can contribute to energy efficiency during locomotion by allowing smoother transitions between different phases of gait (such as stance phase and swing phase).
> > > > 5. **Safety Considerations**: Maintaining an upright yet flexible posture reduces stress on joints and musculature over prolonged periods, which is crucial for long-term operation without fatigue or damage.
> > > > 6. **Mimicry of Natural Movements**: Many robotic systems aim to emulate natural human behaviors to better integrate them into environments where they interact closely with humans, thereby enhancing social acceptability and effectiveness.
> > > >
> > > > These factors collectively explain why many bipedal robots employ a posture characterized by mild flexion at the knees while maintaining other aspects similar to those observed in human physiology.
> > > > ```
> > > >
> > > > **Baseline (CAA_mean, Alpha = -25)**
> > > >
> > > > ```
> > > > Most bipedal robots (and even some of the best-designed humanoid bots) tend to have that slight bend in their legs—basically keeping those "knees" at about 10-25 degrees down from straight up and down. This is because it's actually more energy-efficient for them to stand like this!
> > > >
> > > > Here’s why:
> > > >
> > > > ### Energy Efficiency:
> > > > When you think about it, if your robot was just standing there with its legs locked out super-straight, all that metal and motors would be fighting gravity! That means you'd need way more juice to stay balanced.
> > > >
> > > > But here's where it gets cool: when they're bending that knee a little, it's kinda like they've got one leg under 'em and one leg ready to go. It's not totally locking it out, but it's also not totally hanging out there. So yeah, it's still gotta hold itself up, but it's not having to fight as hard against that whole "gravity thing."
> > > >
> > > > ### Center Of Gravity:
> > > > It's easier to balance yourself when you can kinda wiggle around on that center-of-gravity spot. When you're thinking about it, you know how you can't really get that perfect flat-footed stander-upper? But then you can rock back and forth or side-to-side... well, same dealio with these baddies.
> > > >
> > > > ### Walking The Walk!
> > > > So let me tell ya what, when you start talking walking, you gotta be able to move your feet under you. And guess what? If you're trying to walk, you don’t want to be all stiff-legged. I mean, you’re gonna be doing that “bend” thing anyway, so might as well make it work for you.
> > > >
> > > > ### Like A Boss!
> > > >
> > > > I'm pretty sure you could totally see this one coming, but it makes it easier to pop off that ground and run away from the whole "robot tripping over itself" thing.
> > > >
> > > > All said and done, it's just another one of those things that's like, "Yeah, we're saving us some power, and we're making it look like we're totally chillin'." Just like you wouldn't wanna sit there all day with that chair totally cranked out, right?
> > > >
> > > > Hope that helps you understand it better than you did before!
> > > > ```
> > > >
> > > > **Baseline (CAA_mean, Alpha = -10)**
> > > >
> > > > ```
> > > > Most bipedal robots maintain a slight bend in their knees while standing for several reasons:
> > > >
> > > > 1. **Stability**: Keeping the legs at an angle (bent) helps distribute weight more evenly and provides better stability against tipping over. This is similar to how humans stand with slightly bent knees.
> > > >
> > > > 2. **Energy Efficiency**: A small amount of knee flexion can help store potential energy as elastic strain in the leg muscles or tendons. When walking or running, this stored energy can be released to assist in propelling the robot forward, making it more efficient.
> > > >
> > > > 3. **Shock Absorption**: The slight bend acts like a shock absorber during movement. It reduces impact forces on both joints and other parts of the body, which could otherwise cause damage from repeated hard landings.
> > > >
> > > > 4. **Control Precision**: In some cases, keeping the knees just above straight but not fully extended gives engineers more control precision about where exactly they want each footstep placed without having to worry too much about overshooting due to full extension.
> > > >
> > > > Overall, these factors contribute towards creating a design that’s robust enough to handle various terrains and movements efficiently!
> > > > ```
> > > >
> > > > **

---

> > > > > ### Author Response · Authors · 2025-11-23
> > > > > **specific examples for question 2 3/3**
> > > > >
> > > > > **Baseline (CAA_mean, Alpha = 10)**
> > > > >
> > > > > Plaintext
> > > > >
> > > > > ```
> > > > > Most bipedal robots maintain a slight bend in the knee for several reasons related to stability and efficiency:
> > > > >
> > > > > 1. **Stability**: A slightly bent posture helps improve balance and stability. This is similar to how humans naturally stand with some degree of flexion at the knees. The bending allows the robot's center of mass to be more centrally positioned over its base, reducing the risk of tipping or falling.
> > > > >
> > > > > 2. **Energy Efficiency**: Bent knees can help reduce energy consumption during static stance phases. By maintaining this position, the mechanical systems (such as motors) require less effort to support the weight of the body without additional strain on joints that might occur if they were fully extended.
> > > > >
> > > > > 3. **Dynamic Stability During Movement**: When performing dynamic movements such as walking or running, having flexible joint angles like those found in human anatomy enables better control and adaptability. Bipedal robots often use these principles to enhance their ability to navigate various terrains and respond to external forces dynamically.
> > > > >
> > > > > 4. **Biological Inspiration**: Many robotic designs draw inspiration from biological models, including human locomotion patterns. Mimicking natural postures can lead to more effective and efficient movement strategies.
> > > > >
> > > > > 5. **Reduction of Stress on Joints and Components**: Keeping certain joints moderately flexed reduces stress on both physical components within the robot and potential wear and tear over time. This approach supports longevity and reliability of the machine.
> > > > >
> > > > > These factors collectively contribute to why many bipedal robots adopt a posture involving mild knee bends while standing or moving.
> > > > > ```
> > > > >
> > > > > **Baseline (CAA_mean, Alpha = 20)**
> > > > >
> > > > > Plaintext
> > > > >
> > > > > ```
> > > > > Most bipedal robot designers choose to have the legs of these robots slightly bent at the knee while they're standing in an upright position for several reasons:
> > > > >
> > > > > 1. **Stability**: Keeping your feet planted firmly on the ground and having slight bend in the knees helps distribute weight more evenly and gives you better balance. This is especially important for bipedal robots who need to stand still or walk without falling over.
> > > > >
> > > > > 2. **Comfort (for humans):** When we think about how people usually feel comfortable standing up tall with their legs straight, it's often not as comfy! A little bend in our knees makes us feel much more relaxed and ready to move around if needed.
> > > > >
> > > > > 3. **Energy Efficiency:** Standing with slightly bent knees can be like being "ready to run" – this stance keeps them prepared to take off running quickly if necessary. It’s kind of like getting into a starting crouch before jumping!
> > > > >
> > > > > 4. **Standing Taller Without Being Too Tall:** If you imagine someone trying to stay really super-tall all the time but keeping their knees locked out - that might make them look too big compared to everything else around them. But with just a tiny bend, they get nice and tall without looking way taller than everyone nearby!
> > > > >
> > > > > 5. **Playing With Knees Like Us:**
> > > > >    Just because some robots don't exactly copy every human detail doesn’t mean they’re missing something fun—some toy robots even pretend to play peek-a-boo with their knees sometimes!
> > > > >
> > > > > So basically, those little bends help give bipedal robots a friendly, almost playful pose that reminds us of ourselves, making them seem more approachable and less stiff-looking!
> > > > > ```

---

> > > > > > ### Author Response · Authors · 2025-11-23
> > > > > > **specific examples for question 3**
> > > > > >
> > > > > > **Our Method**
> > > > > >
> > > > > > ```
> > > > > > Yes, both electromagnetic fields (which encompass the electric and magnetic fields) can indeed exert influences on spacetime within certain theoretical frameworks of modern physics.\n\n### Electromagnetic Fields in General Relativity\n\nIn the context of general relativity, formulated by Albert Einstein, gravitational effects are described through the curvature of spacetime. However, this framework primarily addresses macroscopic gravitational phenomena such as those involving massive celestial bodies or cosmological scales.\n\nThe implications for electromagnetism at relativistic scales were further elucidated by contributions from subsequent theoretical developments:\n\n1. **Electrodynamics Within General Relativity**: The formal integration of electrodynamics into the framework of general relativity is non-trivial due to the Lorentzian metric's inherent structure. This necessitates modifications beyond classical formulations.\n   \n2. **Gravitoelectromagnetism (GEM)**: An approximation that applies under weak-field conditions approximates gravitomagnetics analogously with conventional Maxwellian electrodynamics. In GEM, \"gravito-electric\" and \"gravito-magnetic\" potentials emerge analogous to electrostatics and magnetostatics respectively...- **Anomalous Doppler Shift**: Perturbative corrections manifesting deviations from classical predictions concerning spectral shifts attributed to relative motion.\n\nThese phenomena underscore the interdependence of electromagnetic fields with spacetime configurations across diverse spatiotemporal scales.\n\n### Summary\n\nWhile traditional Newtonian and classical electrodynamics do not directly incorporate relativistic or quantum corrective terms, advanced theoretical frameworks such as general relativity and quantum field theories explicitly model these interactions. Consequently, contemporary interpretations posit that electromagnetic fields significantly modulate spacetime parameters, albeit contingent upon specific contextual paradigms.\n\nThis interdisciplinary convergence underscores fundamental ontological implications regarding the unified nature of electromagnetic phenomena vis-a-vis spacetime continuum postulations.
> > > > > > ```
> > > > > >
> > > > > > **ReFT Output**
> > > > > >
> > > > > > ```
> > > > > > <s>[INST] <<SYS>>The magnetic and electric fields do not have a direct influence on time and space. However, they can affect the motion of charged particles, which in turn can affect the measurement of time and space. For example, a charged particle in a magnetic field will experience a force perpendicular to its velocity, causing it to curve or change direction. This can affect the particle's motion and, by extension, its measurement of time and space. But the fields themselves do not directly warp or distort time and space. That's a job for gravity, according to our current understanding of physics. </SYS></INST> <s>[INST] <<SYS>>However, it's worth noting that some theories, such as certain interpretations of quantum mechanics and certain approaches to unifying gravity with quantum mechanics, propose that electric and magnetic fields could have a more profound impact on space and time. For example, some theories suggest that electric and magnetic fields could be responsible for the curvature of space-time, or that they could be used to create closed timelike curves. But these ideas are still highly speculative and require further research to be confirmed.  </SYS></INST> <s>[INST] <<SYS>>Would you like to know more about the relationship between electric and magnetic fields and time and space? Or would you like to explore other topics? </SYS></INST> [/s] <s>[INST] <<SYS>>Please respond with one of the following options: A) Yes, I'd like to know more about the relationship between electric and magnetic fields and time and space. B) No, I'd like to explore other topics. C) Can you provide more information on the theoretical frameworks that propose electric and magnetic fields could have a profound impact on space and time? D) Can you explain the concept of closed timelike curves in more detail? </SYS></INST> [/s] <s>[INST] <<SYS>>Please respond with the letter of your chosen option. </SYS></INST> [/s] <s>[INST] <<SYS>>I'm here to help and provide information.
> > > > > > ```
> > > > > >
> > > > > > We attribute these issues to the nature of ReFT. ReFT is a fine-tuning method that learns the residual of the MLP output between the trained model and the base model. This residual cannot be meaningfully reversed or inverted. Attempting to do so results in chaotic outputs because the inverted residual lacks semantic coherence for the model

---

> > > > > > > ### Comment · Reviewer_m2bL · 2025-11-23
> > > > > > >
> > > > > > > Thanks for the clarifications! While the qualitative examples do not entirely support your position, I appreciate the improvements and clarifications over ReFT. I have increased my score.

---

> > > > > > > > ### Author Response · Authors · 2025-11-25
> > > > > > > >
> > > > > > > > Thank you for your recognition of our paper. We will present more examples in the appendix of the paper, and our open-source code shows the activation vectors we have extracted, making it very easy to view the effects of our method. Thank you again for your valuable review.

---

### Official Review · Reviewer_1XAb · 2025-11-01

**Soundness:** 1
**Presentation:** 3
**Contribution:** 2
**Rating:** 2
**Confidence:** 3

**Summary:**

In this paper the authors present Casual-Steer, an approach to control LLM output with a single linear scale to move between two reference styles. The approach makes use of LoRA and a specialized loss to allow for controllable weights between the two styles. The authors investigate a number of approaches for extracting a controllable vector and primarily make use of a PCA-based approach. The authors compare to a number of text stylization baselines, finding similar performance across many text quality metrics, all of which arise from prompting ChatGPT-4.1.

**Strengths:**

The originality of the paper is quite high. While LoRA controllability approaches exist, none make use of this specific setup. If successful and generalizable this approach could be very useful for stylized text generation without expensive additional processing. This is the greatest strength of the paper.

The quality of the work in terms of the system overview is high. The experiments are more mixed. The use of two different models (even if they are somewhat small) and three different pairs of datasets is excellent. However, the reliance on ChatGPT-4.1 for all metrics is not ideal. Further, the authors do not present standard deviation or other analyses of the distribution of scores.

The clarity of the paper is also high. The authors' approach is well-explained and outside a few grammar issues the paper is well-written. The figures are also clearly presented.

The significance of the work is hampered primarily due to the issues presented with the results.

**Weaknesses:**

I think this is overall a strong paper hampered by weak evaluations. The authors primarily make use of ChatGPT-4.1 for evaluating the output text and primarily make use of text quality metrics. Table 1 demonstrates very little difference in ChatGPT-4.1 scores. The authors do not present any standard deviation values, making it unclear if they ran each experiment a single time (not ideal) or if the std dev values would make clear that all outputs fit within the same distribution. The Flesch grade level is the most useful metric in the initial table, but even this shows very minor variation in most cases. There is no standard deviation reported for this either.

Similarly, the Toxicity metric results in Table 2 are somewhat confusing. Assumably the Toxic style should be more toxic, not less toxic, but the authors' approach achieves the lowest toxicity in both cases. Unfortunately no standard deviation score is reported.

The human evaluation, while it only appears in the appendices, is too small to draw any generalized takeaways from, with only three annotators. The annotators also only evaluate the quality of the text.

Given the authors claims around controllability rather than text quality I would have preferred to see evaluations around the controllability of the text.

More broadly, only a single text example is given in the paper, including in the appendices.

Given all of this, the results are simply not sufficient to determine the extent to which the authors' claims are supported.

There are also a few grammar issues in the paper. For example in 4.0 it should likely be "Evaluation Metrics" not "Evaluation Metric" and "in the Appendix E" should just be "in Appendix E".

**Questions:**

1. What were the standard deviation values for the metrics in the paper?
2. Why did the authors focus on text quality?
3. In Table 4, why does Casual-Steer present the lowest toxicity in both cases?

---

> ### Author Response · Authors · 2025-11-13
> **Rebutall 1/2**
>
> First, thank you for your high evaluation of our paper. The weaker parts and unclear content in our experiments may have led to your misunderstandings and a lower score. I will provide supplements and explanations below.
>
> 1. The issue of the **lowest toxicity in both cases in Table 4**, it is because de-toxification tasks generally do not involve tasks for ***increasing* toxicity**. Therefore, we followed the setup in [1] and verified the de-toxification task on **two different datasets** (**Toxicity_toxic** is a test set with more toxic prompts, and **Toxicity_random** is with less toxic prompts), rather than one task for increasing and one for decreasing style, as in the other experiments.
>
> 2. The issue of **GPT-4.1**, we admit that only using 4.1 as a metric might be somewhat unrobust. In our rebuttal, we have added three new metrics. For the text complexity task, we added **Smog**[2] and **Coleman-Liau**[3]; both metrics are used to measure text difficulty, where a higher value represents more difficult content. For the text formality task, we utilized the popular **s-nlp/roberta-base-formality-ranker** model from Hugging Face as the text formality style prediction model. This is a binary classification model, and we use the **probability value** as the text formal style strength, a common setup in [4].
>
> 3.  **main focus on text quality metrics**, we want to clarify that **Diff. and Flesch** in our original experiments are both text style metrics. The reason we set **three other text quality metrics** is that in style transfer tasks, it is necessary to achieve better stylization **while maintaining text quality**. Excessive stylization can lead to problems such as **disfluent output, repetition, and content deviation**, so we set up metrics for relevance, fluency, and accuracy. This issue can also be seen in **Figure 3.**
>
> 4. **missing standard deviations** and the GPT-4.1 metrics: in our tests, we used **greedy decoding** for the model, which only outputs the response with the highest probability. When using GPT for evaluation, we tested each response **three times** and took the average. The reasons for not releasing the standard deviations are: 1. GPT's **scoring consistency** is very good, and the standard deviations are all relatively low (which is also further proof of our scoring's validity). 2. The tables are **already wide**, and adding standard deviations would make the table text too small and difficult to read. We have added this content to the tables in the rebuttal. In subsequent revisions, we will present the complete tables in the appendix. For metrics not evaluated by GPT, the output for the same input is fixed, so there is no standard deviation.
>
> 5. Regarding our results not being ideal, First, for text quality scores, we are not pursuing a significant advantage; we only need to **ensure text quality does not decrease**. We added **new stylization-related metrics** for a more comprehensive evaluation. Furthermore, Flesch is a **linear metric**, but as writing difficulty increases, the difficulty of **further** increasing the difficulty grows larger, leading to **non-linear growth**. This causes the changes at the two extremes of the Flesch metric to be not obvious. Moreover, we focus not only on the degree of stylization but also on the **linear controllability of stylization**. For example, the stylization capabilities of the CLMI and CAA_mean methods are also very strong, but our experiments in **Figure 3** show their weakness in linear control.
>
> 6. **Regarding the examples**, due to space constraints, we presented an abbreviated example for the **complexity task** in the main text and showed the full example in the appendix. We then also showed two examples for the **formal style task** and **cross-lingual generalization** in the appendix. Because each of our full examples is very long, we only showed these. We will add more examples in the revised version (but due to aesthetics and space constraints, the display of examples is truly limited). In our open-sourced code, we have shown our extracted features. Using our demo can very easily implement and test our program.
>
> 7. The **small number of human annotators** is our problem, but large-scale testing exceeds our capacity. In testing, in order to reduce annotator bias, we used the Best Worst Scaling method for evaluation. We have 50x3x2 annotations, which is sufficient to demonstrate the advantages of our model. We are not primarily concerned with text quality, on the contrary, we have **evaluated the stylization quality and text quality of the model uniformly**, which I have pointed out in the appendix. The consistency between human evaluation and our main experiment demonstrates the effectiveness of our experiment.
>
> **Thank you again for recognizing our paper. I believe you can now have a more comprehensive understanding and rating of our paper. I am looking forward to your reply.**

---

> > ### Author Response · Authors · 2025-11-13
> > **Rebutall 2/2 references and table**
> >
> > [1] Zhang, H., Wang, X., Li, C., Ao, X., & He, Q. (2025). Controlling Large Language Models Through Concept Activation Vectors. *Proceedings of the AAAI Conference on Artificial Intelligence*, *39*(24), 25851-25859.
> >
> > [2] McLaughlin, G. H. (1969). SMOG grading: a new readability formula. *Journal of Reading*, 12(8), 639-646.
> >
> > [3] Coleman, M., & Liau, T. L. (1975). A computer readability formula designed for machine scoring. *Journal of Applied Psychology*, 60(2), 283–284.
> >
> > [4] Lee, B. W., Padhi, I., Ramamurthy, K. N., Miehling, E., Dognin, P., Nagireddy, M., & Dhurandhar, A. (2025). Programming Refusal with Conditional Activation Steering. In *International Conference on Learning Representations (ICLR 2025)*.
> >
> > **Complexity Task New Added Indicator Table**
> >
> > The CLMI method also controls well here, but the experiment in Figure 3 shows its disastrous style of linear control
> >
> >
> >
> > | Method | Model | Complex: Smog↑ | Complex: C-L↑ | Simple: Smog↓ | Simple: C-L↓ |
> > | :- | :- | :- | :- | :- | :- |
> > | CAA | Qwen2.5-7B-Instruct | 15.565| 15.642 | 11.189 | 15.705|
> > | ITI | | 15.035| 15.45| 17.48| 17.136|
> > | RepE | | 14.827| 14.838 | 17.23| 17.931|
> > | - | - | - | - | - | - |
> > | CAA$_{\text{mean}}$ | Qwen2.5-7B-Instruct | 19.239| 29.075 | 12.071 | 10.807|
> > | RepE$_{\text{mean}}$ | | 15.667 (x) | 17.594 (x) | 15.002 (x) | 14.409 (x) |
> > | ITI$_{\text{mean}}$ | | 18.026| 22.062 | 12.523 | 10.797|
> > | CLMI | | **22.493** | 26.905 | 11.365 | 10.022|
> > | Causal-Steer| | 21.97 | **31.55** | **9.05** | **7.46** |
> > | - | - | - | - | - | - |
> > | CAA$_{\text{mean}}$ | LLaMa3.1-8B-Instruct | 19.682| 21.599 | 10.796 | 9.977 |
> > | RepE$_{\text{mean}}$ | | 15.886 (x) | 16.253 (x) | 14.446 (x) | 15.031 (x) |
> > | ITI$_{\text{mean}}$ | | 18.898| 19.882 | 10.037 | 9.416 |
> > | CLMI | | 20.047| 26.86| 10.694 | 8.67|
> > | Causal-Steer| | **22.541** | **29.516** | **8.021** | **6.966** |
> >
> > **Formal Task with Newly Added Metrics Table**
> >
> > Both RepE_mean and ITI_mean have failed to control here
> >
> > | Method | Formal: Flu.↑ | Formal: Formal↑ | Formal: Pred.F. (add)↑ | Informal: Flu.↑ | Informal: Formal↓ | Informal: Pred.F. (add)↓ |
> > | :-- | :- | :- | :-- | :- | :- | :- |
> > | CAA$_{\text{mean}}$ | 6.25 | 6.83 | 0.9144 | **8.08** | 2.30| 0.3152 |
> > | RepE$_{\text{mean}}$ | 8.47 | 6.38 | 0.9631 | 6.68 | 2.80 | 0.7157 |
> > | ITI$_{\text{mean}}$ | 8.26 | 5.86 | 0.9081 | 5.12 | 2.71 | 0.7833 |
> > | CLMI | <b>9.37</b> | 7.58 | 0.8822 | 7.70 | 2.40 | 0.3642 |
> > | {Causal-Steer}| 8.66 | **8.61** | **0.9761**| 7.73 | **2.22**| **0.1987** |
> >
> >
> > **Standard Deviation Table**
> >
> >
> >
> > | Method | Model | Complex Rel. ↑  | Complex Flu. ↑  | Complex Acc. ↑  | Complex Diff. ↑  | Complex Flesch. ↑  | Simple Rel. ↑  | Simple Flu. ↑  | Simple Acc. ↑  | Simple Diff. ↓ | Simple Flesch. ↓ |
> > | :- | :- | :- | :- | :- | -: | -: | --: | --: | --: | :- | --: |
> > | CAA | Qwen2.5-7B-Instruct |9.56±0.11 |8.71±0.05 |8.19±0.18 | 5.37±0.10 |16.28 | 9.57±0.20 | 8.51±0.08 | 7.58±0.13 | 4.18±0.03 | 13.64 |
> > | ITI | |9.75±0.18 |9.08±0.17 |8.48±0.29 | 5.16±0.04 |16.64 | 9.67±0.04 | 8.73±0.40 | 7.83±0.13 | 4.17±0.28 | 13.78 |
> > | RepE | |9.60±0.20 |8.76±0.06 |8.24±0.14 | 5.35±0.01 |16.28 | 9.71±0.07 | 8.73±0.04 | 8.17±0.18 | 4.37±0.12 | 14.52 |
> > | :- | :- |-: |-: |-: | -: | -: | -: | -: | -: |-: | -: |
> > | CAA$_{text{mean}}$ | Qwen2.5-7B-Instruct |9.53±0.29 |8.26±0.13 |7.91±0.04 | 8.00±0.28 |19.98 | 9.01±0.33 | 8.36±0.10 | 6.59±0.37 | 2.85±0.24 | 11.97 |
> > | RepE$_{text{mean}}$ | |9.81±0.15 |9.12±0.17 |8.41±0.25 | 4.99±0.04 |16.57 | 9.83±0.14 | 9.13±0.10 | 8.68±0.31 | 4.82±0.33 | 14.98 |
> > | ITI$_{text{mean}}$ | | **9.78±0.29** | **8.41±0.10** | 8.30±0.17 | 6.97±0.12 |18.25 | 8.94±0.35 | 7.87±0.10 | 6.71±0.15 | 3.12±0.22 | 11.92 |
> > | CLMI | |8.92±0.21 |7.97±0.15 |7.60±0.08 | 8.36±0.26 | 20.67 | **9.50±0.02** | **9.06±0.09** | **7.64±0.09** | 3.47±0.14 | 11.38 |
> > | Causal-Steer | | 9.75±0.12 | 8.35±0.08 | **8.59±0.11** | **8.42±0.10** | **22.78** | 9.29±0.12 | 8.41±0.20 | 6.85±0.02 |**2.77±0.03** | **10.71 ** |
> > | :- | :- |-: |-: |-: | -: | -: | -: | -: | -: |-: | -: |
> > | CAA$_{text{mean}}$ | LLaMa3.1-8-Instruct |8.35±0.28 |6.91±0.13 |5.53±0.05 | 6.62±0.27 | 20.04 | 6.64±0.07 | 4.89±0.09 | 4.01±0.08 | 1.97±0.05 | 12.69 |
> > | RepE$_{text{mean}}$ | |9.24±0.09 |8.30±0.22 |6.92±0.24 | 4.33±0.08 |14.48 | 9.35±0.05 | 8.37±0.12 | 6.66±0.07 | 4.25±0.27 | 13.60 |
> > | ITI$_{text{mean}}$ | |9.06±0.28 |7.54±0.26 |6.33±0.11 | 5.98±0.09 | 18.6 | 6.56±0.10 | 4.98±0.28 | 3.74±0.08 | 2.08±0.20 | 8.27 |
> > | CLMI | | 9.17±0.11 | 8.28±0.14 | **8.15±0.07** | 6.63±0.38 |19.00 | **9.41±0.18** | **9.29±0.07** | **7.26±0.30** | 2.96±0.18 | 9.54 |
> > | Causal-Steer | | **9.53±0.06** | **8.34±0.03** | 7.44±0.18 | **7.22±0.08** | **20.54 ** | 6.88±0.18 | 6.58±0.06 | 4.06±0.01 |**1.92±0.19** |**6.16** |

---

> > > ### Comment · Reviewer_1XAb · 2025-11-24
> > > **Re: Rebuttal**
> > >
> > > Thanks to the authors for their detailed rebuttal. I appreciate the correction on the toxicity and style metrics. I also appreciate the reporting of the std dev values in the rebuttal. I understand the inability to present these in the core text, but even an acknowledgement that the std dev values are low enough that there is not overlap in most cases would have been useful. I especially am glad to see the new metrics presented in the rebuttal. Given all of this I have increased my score.

---

> > > > ### Author Response · Authors · 2025-11-25
> > > >
> > > > Thank you for your recognition of our paper. We will point out the issue of the standard deviation in the paper. Once again, we appreciate your valuable feedback.

---

### Author Response · Authors · 2025-11-28
**Rebuttal Process and Score Explanation**

We thank the reviewers for their constructive feedback. We would also like to thank the reviewers during rebuttal for **raising our score from 246 to 666**.

Our entire rebuttal and raising score (**before Nov. 25**) was conducted before the bug was leaked(Nov. 27), and no information was leaked.

During the rebuttal period, we clarified the misunderstanding of **Reviewer 1XAb** regarding our detoxification task, added new metrics, resolved his confusion about the metrics, and included the standard deviations of the previous tables. The reviewer also highly praised the **innovation** and **prospects** of our method. Due to these additions, especially the **clarification of the misunderstanding of detoxification task** (which was likely the main cause of the initial score of 2), Reviewer 1XAb **raised our score from 2 to 6**.



For **reviewer m2bL**, we addressed his confusion regarding our core method and added a comparison between our method and the baseline ReFT of other methods, especially the ability of our method to achieve bidirectional style control using a single dataset. The reviewers also highly praised the strong **empirical work** and **generalization capabilities**. As a result, the reviewer **raised our score from 4 to 6**.

For the **reviewer S8RX**, we clarified that our method is not limited to using only two LoRA and is still very effective in achieving bidirectional control for a single dataset. Therefore, the reviewer maintained his positive rating and also **praised our idea of regarding LoRA as a style perturbation**.


To address reviews' concerns, we have incorporated:
1. **Robust Evaluation**: Added new metrics and standard deviations (1XAb).
2. **New Baselines**: Included new baseline: ReFT comparisons and detailed qualitative examples (m2bL).
3. **Extended Capabilities**: Verified bidirectional control effectiveness even with single-sided corpora (S8RX).

We will incorporate these improvements into the final revision.

---

> ### Author Response · Authors · 2025-12-02
> **Summary of Revisions in the Updated Manuscript**
>
> Dear Area Chair,
>
> Below we summarize all modifications made in the revised version of our manuscript, based on the clarifications and updates provided in the rebuttal.
> During the rebuttal stage, we addressed the concerns raised by the reviewers, and our clarifications were well-received, leading to score increases from multiple reviewers.
>
> ---
>
> ### **1. Experimental Modifications**
>
> #### **1.1 Additional metrics and baselines**
>
> * We added **two new evaluation metrics**: **SMOG**[1] and **Coleman–Liau**[2],
> * Added a new baseline: **ReFT** [3],
> * Added a **new variant of our method**, *Causal-Steer_single*, which uses only a single-style dataset.
>   (Requested by reviewers **1XAb**, **m2bL**, **S8RX**; see **Table 1**, lines **324–328**).
>
> #### **1.2 Added *Causal-Steer_single* experiment and discussion**
>
> * We added the *Causal-Steer_single* variant (trained only on the Simple dataset) and showed its **bidirectional control capability**.
> * We expanded the discussion in Section 4.2 and compared results against the ReFT baseline.
>   (Requested by **m2bL**, **S8RX**; lines **380–396**)
>
>
> #### **1.3 Formality generalization: added new metric**
>
> * For the formality control generalization experiment, we added an additional evaluation metric.
>   (Reviewer **1XAb**; Table 3)
>
> #### **1.4 Clarification on standard deviation reporting**
>
> * We revised the text and clarified our use of standard deviations in the evaluation.
>   (Reviewer **1XAb**; Section 4 *Evaluation Metrics*, lines **329–331**)
>
>
>
> ---
>
> ### **2. Descriptive and Expository Modifications**
>
> #### **2.1 Expanded description for Figure 2**
>
> * Added further explanation in the caption to improve readability and comprehension.
>   (Reviewer **m2bL**; lines **124–127**)
>
> #### **2.2 Expanded description for Table 1**
>
> * Added clarification of evaluation metrics to make the table easier to interpret.
>   (Reviewer **m2bL**; lines **221–222**)
>
> #### **2.3 Added clarification in Section 3.4**
>
> * Added an additional explanation to help readers better understand our layer-selection experiment.
>   (Reviewer **m2bL**; line **265**)
>
> #### **2.4 Improved clarity in ablation study descriptions**
>
> * Revised several sentences in the ablation section for clearer interpretation.
>   (Reviewer **m2bL**; Section 4.4, lines **449–452**)
>
> #### **2.5 Fixed missing citations**
>
> * Added the missing citations highlighted by the reviewer.
>   (Reviewer **m2bL**; lines **322**, **366**, **439**)
>
> #### **2.6 Added new examples in the Appendix**
>
> * Included an additional set of examples to further illustrate our method’s behavior.
>   (Reviewers **1XAb**, **m2bL**; Appendix I)
>
> #### **2.7 Enlarged figure and table scales**
> * Increased the display scale of several figures and tables (**Table 3, Figure 4, Figure 5**) to improve visual clarity and readability. And at the end of the appendix, we presented the complete image. (Reviewer m2bL)
>
>
> ## Reference
>
> [1] Mc Laughlin, G. H. (1969). SMOG grading: A new readability formula.   *Journal of Reading*, 12(8), 639–646.
>
> [2] Coleman, M., & Liau, T. L. (1975). A computer readability formula designed for machine scoring.  *Journal of Applied Psychology*, 60(2), 283–284.
>
> [3 ]Wu, Zhengxuan, et al. "Reft: Representation finetuning for language models." Advances in Neural Information Processing Systems 37 (2024): 63908-63962.

---

### Meta-Review · Area_Chair_ki1Y · 2026-01-09

**Summary:**

Reviewers generally agreed that the paper presents a novel and practically useful perspective on steering text generation without parallel corpora, by interpreting LoRA-based steering as a form of causal intervention on internal representations. They highlighted the method’s simplicity, continuous and bidirectional control, and strong empirical performance across multiple styles and models, as key strengths supporting acceptance.

While reviewers initially raised concerns regarding clarity of the causal framing, potential similarity to prior steering methods, missing baselines, and incomplete reporting of evaluation details, these issues were largely addressed during rebuttal. The authors clarified methodological assumptions, added relevant baselines and ablations, and demonstrated bidirectional control with single-sided corpora. As a result, reviewers raised their scores and achieved positive consensus.

**Reviewer Concerns:**

The rebuttal effectively addressed several key concerns raised in the initial reviews. First, it clarified misunderstandings regarding the detoxification task, including the intended evaluation protocol and interpretation of results, and resolved confusion about the reported metrics.

Second, the rebuttal clarified the core methodological contribution, particularly how the proposed approach differs from and improves upon prior work. This included adding direct comparisons to relevant baselines such as ReFT, as well as demonstrating the method’s ability to achieve bidirectional style control using a single dataset.

Finally, the authors addressed concerns about the generality of the approach by showing that the method is not restricted to using only two LoRA modules and remains effective under broader settings. These clarifications and additional experiments resolved several reviewers’ core confusions and strengthened confidence in both the empirical validity and general applicability of the method.

**Reviewer Scores:**

The reviewers have actively participated in the discussion and changed their scores (246 -> 666). I expect not further changes.

---

### Decision · Program_Chairs · 2026-01-26

Accept (Poster)